# Non-volatile magnon transport in a single domain multiferroic

Sajid Husain [1,19] ✉, Isaac Harris [1,2,19], Peter Meisenheimer [3], Sukriti Mantri [4], Xinyan Li [5], Maya Ramesh [6], Piush Behera [1,3], Hossein Taghinejad [2,7], Jaegyu Kim [3], Pravin Kavle [1,3], Shiyu Zhou [8], Tae Yeon Kim [3], Hongrui Zhang [1,3], Paul Stevenson [9], James G. Analytis [2], Darrell Schlom [6], Sayeef Salahuddin [3,10], Jorge Íñiguez-González [11,12], Bin Xu [13], Lane W. Martin [1,3,5,14,15], Lucas Caretta [16], Yimo Han [5], Laurent Bellaiche [4,17], Zhi Yao [18] ✉ & Ramamoorthy Ramesh [1,2,3,5,14,15] ✉

Antiferromagnets have attracted significant attention in the field of magnonics, as promising candidates for ultralow-energy carriers for information transfer for future computing. The role of crystalline orientation distribution on magnon transport has received very little attention. In multiferroics such as $BiFeO_3$ the coupling between antiferromagnetic and polar order imposes yet another boundary condition on spin transport. Thus, understanding the fundamentals of spin transport in such systems requires a single domain, a single crystal. We show that through Lanthanum (La) substitution, a single ferroelectric domain can be engineered with a stable, single-variant spin cycloid, controllable by an electric field. The spin transport in such a single domain displays a strong anisotropy, arising from the underlying spin cycloid lattice. Our work shows a pathway to understanding the fundamental origins of magnon transport in such a single domain multiferroic.

Electromagnetic coupling offers a foundational framework for transforming between magnetic and electric fields, primarily facilitated by the principle of magnetic induction through electric currents[1, 2]. For applications such as manipulating the magnetization of nanoscale magnets in integrated memory and logic, however, the conventional Oersted field approach has been proven to be energy-inefficient and impractical[3]. To address the imperative of low-energy consumption in nonvolatile magnetic memory and logic, a promising new avenue has emerged − direct voltage control of magnetism[4–9]. In multiferroics, due to the non-centrosymmetric crystal structure (with strong

[1]Materials Science Division, Lawrence Berkeley National Laboratory, Berkeley, CA, USA. [2]Department of Physics, University of California, Berkeley, CA, USA. [3]Department of Materials Science and Engineering, University of California, Berkeley, CA, USA. [4]Smart Ferroic Materials Center, Physics Department and Institute for Nanoscience and Engineering, University of Arkansas, Fayetteville, Arkansas, USA. [5]Materials Science and NanoEngineering, Rice University, Houston, Texas, USA. [6]Department of Materials Science and Engineering, Cornell University, Ithaca, NY, USA. [7]Heising-Simons Junior Fellow, Kavli Energy NanoScience Institute (ENSI), University of California, Berkeley, CA, USA. [8]Department of Physics, Brown University, Providence, RI, USA. [9]Department of Physics, Northeastern University, Boston, MA, USA. [10]Department of Electrical Engineering and Computer Sciences, University of California, Berkeley, CA, USA. [11]Department of Materials Research and Technology, Luxembourg Institute of Science and Technology, Esch/Alzette, Luxembourg. [12]Department of Physics and Materials Science, University of Luxembourg, Belvaux, Luxembourg. [13]Jiangsu Key Laboratory of Frontier Material Physics and Devices, School of Physical Science and Technology, Soochow University, Suzhou, China. [14]Departments of Chemistry, and Physics and Astronomy, Rice University, Houston, TX, USA. [15]Rice Advanced Materials Institute, Rice University, Houston, TX, USA. [16]School of Engineering, Brown University, Providence, RI, USA. [17]Department of Materials Science and Engineering, Tel Aviv University, Ramat Aviv, Tel Aviv 6997801, Israel. [18]Applied Mathematics and Computational Research Division, Lawrence Berkeley National Laboratory, Berkeley, CA, USA. [19]These authors contributed equally: Sajid Husain, Isaac Harris. ✉e-mail: shusain@lbl.gov; jackie-zhiyao@lbl.gov; rramesh@berkeley.edu

magnetic anisotropy), the space inversion and time reversal symmetries are simultaneously broken. This leads to the foundation of the ferroelectricity and magnetoelectric effect in multiferroics. Recent proposals use the magnetoelectric coupling inherent in some multiferroics, which allows for direct electric field control of the magnetic state in such a material[7, 10]. A notable example of this innovation is the magneto-electric spin-orbit (MESO) logic device structure, proposed as an inherently non-volatile substitute for complementary metal-oxide-semiconductor (CMOS) devices in integrated logic-in-memory applications[11, 12]. To this end, BiFeO$_3$, possessing strong antiferromagnetic magnetoelectric coupling[10, 13, 14], is considered a desirable material for MESO-type devices. In addition, due to their antiferromagnetic character, the materials are robust against external magnetic fields and possess potentially faster-switching dynamics than ferromagnets. Recently, it has also been shown to be an efficient system for demonstrating switchable magnon spin currents[15, 16]. Under an external magnetic field, additional contributions such as the Nernst effect and the anomalous Nernst effect can occur (reported in Supplementary Information of ref. 17). However, controlling magnetization through an electric field eliminates these contributions. This approach offers a detailed understanding and tunability of polar and magnetic order parameters in multiferroics. This electric field switchable electro-magnon coupling allows for a simplified version of the MESO device i.e., the antiferromagnetic state is directly non-volatile read out using the spin-orbit metal in direct contact with the antiferromagnetic layer, i.e., without an interleaving ferromagnetic layer. In this context, there have been attempts to realize antiferromagnetic state readout using the electrical control of magnon transport in BiFeO$_3$ (refs. 17, 18). However, challenges remain due to the existence of ferroelastic domain walls, which can inhibit magnon transport through a diffusive nature, as well as a large electric field required to switch the polarization. These issues necessitate further innovations to achieve efficient magnon output. To address this, La -substitution in BiFeO$_3$ has been proposed to significantly reduce the switching field[19]. The open question remains: How can we uncover methods to improve performance magnitude and deepen our understanding of magnon transport in La-substituted BiFeO$_3$? Addressing these questions has the potential to unlock the application-oriented significance of these materials for broader future problems.

## Results

The ground state of bulk BiFeO$_3$ has a large polarization ($\sim 90\,\mu C/cm^2$) along $[111]_{pc}$ (pc: pseudocubic) and exhibits a canted G-type antiferromagnetism modulated by a spin cycloid (period $\sim 65$ nm due to the inverse spin current effect[20]) below the Néel temperature (640 K). Rhombohedral BiFeO$_3$ in its G-type antiferromagnetic state shows Rashba splitting which is intrinsically linked to spin-dependent transport[21]. BiFeO$_3$ features two principal Dzyaloshisnkii-Moriya (DM)-like interactions, linked to the polarization and the antiferrodistortive octahedral tilts[22], where the tilts and polarization are strongly coupled[23–25]. The octahedral tilt induces a weak magnetic perturbation and corresponding spin density wave on top of the antiferromagnetic cycloid of BiFeO$_3$[22, 25]. This can be imaged directly using scanning Nitrogen-vacancy (NV) magnetometry[26]. To introduce tunability in multiferroic properties, rare earth substitution has shown great potential. Often, in these systems, the ferroelectric polarization moves away from $[111]_{pc}$ (hereafter all directions are used in the pseudocubic notation unless otherwise specified)[27, 28] introducing competition between ferroelectric and antiferroelectric phases[19, 28–31]. This may allow for additional switching pathways compared to the parent compound BiFeO$_3$, leading to the possibility for new ferroelectric domain configrations. Understanding the formation of a single-domain multiferroic and its potential as a model system for efficient spin magnon transport is the focus of this work.

Theoretical calculations predict a cycloidal magnetic ground state in BiFeO$_3$, illustrated in Fig. 1a. La-substitution modifies the structure and impacts both the magnitude and direction of the spontaneous polarization significantly, which is observed to be along [112] and is $\sim 50\%$ smaller than BiFeO$_3$. This agrees with experimental values and is supported by high-resolution polar maps (Fig. 1a, b and Supplementary Note 2). The fundamental origins of spontaneous polarization (its magnitude and direction) arise from the electronic structure of the Bi-ion in BiFeO$_3$. As such, $\sim 90\%$ of the spontaneous polarization arises from the 6s electrons in the Bi-ion[32]. Upon substituting Bi with La (which does not have any outer shell electrons, unlike Bi$^{+3}$), one is effectively progressively removing the 6s electrons, and thus the polarizability of the La-substituted BiFeO$_3$ is decreased. Indeed, beyond $\sim 18\%$ of La substitution, the material undergoes a phase transition from a polar state to an antipolar phase[19]. The reduction in spontaneous polarization is accompanied by a corresponding reduction in the polarization-dependent DM interaction strength[33], and thus the cycloid becomes less energetically stable. In other words, reducing polarization ($P$) enhances the tilting, and consequently, the tilt-induced-canting of the magnetization becomes larger. The coupling between octahedral tilt and spontaneous polarization with and without La-substitution has been studied in multiple publications, the most recent being an ab initio study, by Fedorova et al.[23] and experimentally in refs. 19, 27, 28. These findings confirm that La-substitution modifies the energy landscape for both the ferroelectric and antiferromagnetic states in BiFeO$_3$ (Fig. 1b). In the case of pure BiFeO$_3$, the polar structure is $R3c$, and the cycloid is a stable magnetic state. Interestingly, with the 15% lanthanum substitution ("Methods"), the uniform canted moment state ($M$1 and $M$2, Methods) becomes closer in energy to the cycloid state (Fig. 1a). This would favor the transition from a spin cycloid state to a complex state upon La-substitution.

In this spirit, Bi$_{0.85}$La$_{0.15}$FeO$_3$ films have been deposited on $(110)_O$ (O: orthorhombic) DyScO$_3$ substrates ("Methods") (Supplementary Notes 1, 2). The La (15%) (in this study) was chosen due to its lowest coercive field, and past work has shown that above $\sim 18\%$ La, a non-polar antiferroelectric phase emerges[34]. Therefore, our interest is to maintain a single-domain ferroelectric phase and manipulate the polarization magnitude/direction and the resultant domain structure using 15% La substitution. Using piezo-force microscopy (PFM) and NV magnetometry, as predicted (Fig. 1a, b), the pure cycloid (within 71° ferroelectric BiFeO$_3$) and mixed state of cycloid+canted antiferromagnet phase(in blocky-mixed ferroelectric Bi$_{0.85}$La$_{0.15}$FeO$_3$) are both observed in a mixed equilibrium state (Fig. 1e–h). Note that the large canted antiferromagnetic phase is expected to generate a larger magnetic field, as is evident from the full-B (Supplementary Fig. 16c) in comparison to the pure cycloid phase (Supplementary Fig. 16f). This supports our theoretical assumptions of polarization and DMI relationship based on the effective Hamiltonian. To then understand the effect of electric field on the as-grown ferroelectric domain structure, and therefore the ferroelectric polarization, in-plane capacitors were fabricated by optical lithography (ex-situ sputtered platinum (Pt) wires $120\,\mu m \times 1.3\,\mu m \times 15$ nm, with $\sim 2\,\mu m$ spacing and resistivity of $\sim 20\,\mu\Omega$ cm). The devices were patterned along four different angles in which the long-axis of Pt electrode pairs are parallel to the substrate [100], [010], [110], and [$\bar{1}$10] pseudocubic directions (Fig. 2a). To visualize the ferroelectric domain reversal across the in-plane devices (Fig. 2b, $P$ vs $E$ hysteresis), PFM images were recorded after poling in two opposite electric field directions (Figs. 2c, d and 3). For a field applied along the [100] direction, in-plane poling leads to the formation of a single ferroelectric domain (discussed later), which is the novel feature of Bi$_{0.85}$La$_{0.15}$FeO$_3$. This has a powerful impact on the magnetic cycloid, which is particularly important for spin transport (discussed later). The formation of a single ferroelectric domain is further verified by rotating the device and performing PFM imaging (Supplementary Note 3), which shows the uniform domain contrast

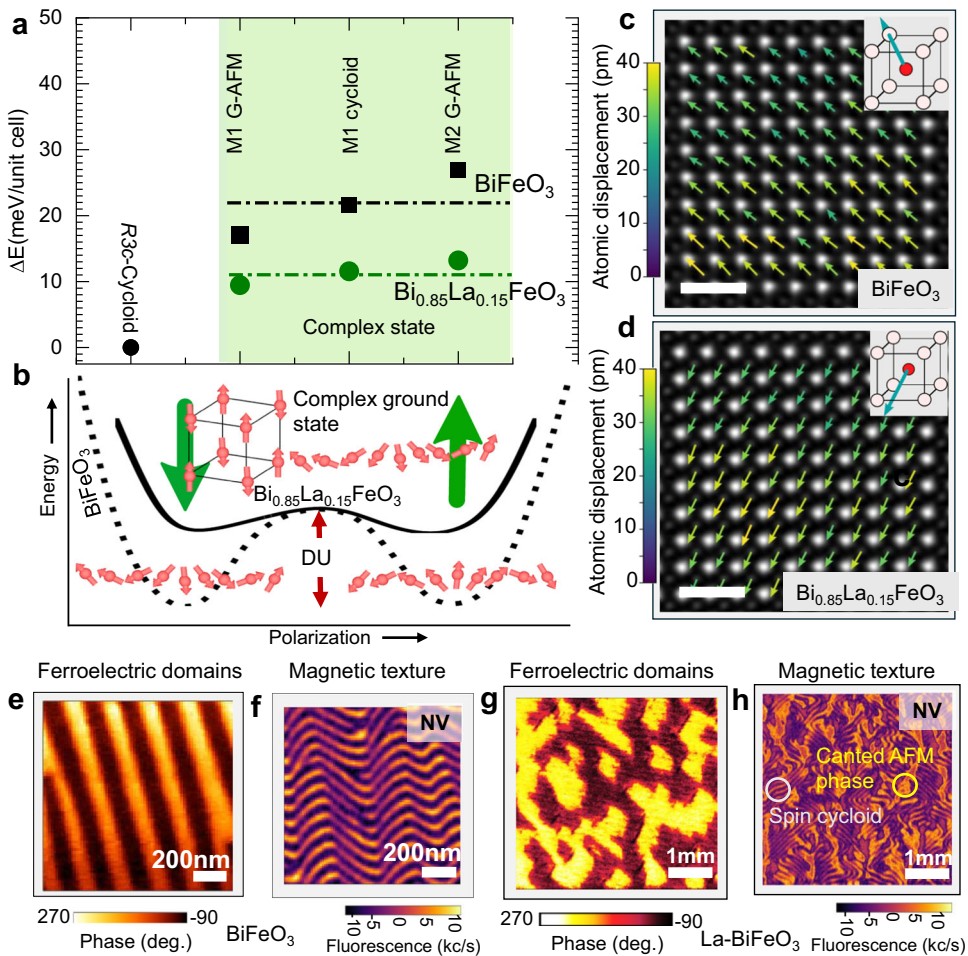

**Fig. 1 | Ferroelectric and magnetic ground state of La substituted BiFeO₃.**
Effective Hamiltonian calculated (**a**) magnetic ground state energy of the spin cycloid and G-type antiferromagnetic phase in BiFeO₃ and Bi₀.₈₅La₀.₁₅FeO₃. *R3c* represents uniform, and *M*1 and *M*2 represent modulated polar configurations ("Methods"). A spin cycloid exists in the ground state of BiFeO₃ whereas a complex mixed state becomes increasingly stable in Bi₀.₈₅La₀.₁₅FeO₃ thin films due to the decreasing energy difference between the two magnetic configurations. Complex state (shaded area) is defined as the mixed state of the spin cycloid and G-type canted antiferromagnetic phase. The dotted lines represent the average energy of the two systems in the complex state configuration. **b** Schematic of the energy landscape of the BiFeO₃ and Bi₀.₈₅La₀.₁₅FeO₃ where the ground state of magnetic textures such as G-type antiferromagnet and spin cycloid phases in the two systems is described. Red arrows form the spin cycloid in the ground state of BiFeO₃ with

the $\Delta U$ energy barrier, whereas the complex state is formed in Bi₀.₈₅La₀.₁₅FeO₃ due to the reduced energy barrier on La substitution. Green up/down arrows represent the polarization in the double well structure. **c, d** High angle annular dark field (HAADF) scanning transmission electron microscopy (STEM) images and polar vector mapping in BiFeO₃ and Bi₀.₈₅La₀.₁₅FeO₃. Insets are the schematics of the estimated polarization direction in the unit cell of BiFeO₃ and Bi₀.₈₅La₀.₁₅FeO₃. Colored arrows indicate the direction as well as the magnitude of the atomic displacement vector vis-à-vis polarization. The average polarization is no longer along [111] after La-substitution. **e–h** Ferroelectric domain (PFM) and corresponding magnetic texture (iso-B NV images) of BiFeO₃/Bi₀.₈₅La₀.₁₅FeO₃ in the pristine state. In (**h**), two types of contrast are visible: the stripe-like contrast from the spin cycloid phase, and the more uniform contrast from a canted antiferromagnetic (AFM) phase.

---

indicative of a single ferroelectric domain. Previously, monodomain features were realized through a non-trivial approach in BiFeO₃ using a scanning probe-tip-based method in slow scan mode to physically write a monodomain using a localized in-plane electric field from the scanning probe tip[26, 35], requiring time (several minutes per micron) and an extremely careful experimental protocol[36, 37], compared to the direct voltage pulse induced switching of La substituted BiFeO₃ single domain that we have adopted in this work.

In the case of Bi₀.₈₅La₀.₁₅FeO₃, the polarization is deterministically switched by an electric field at the macroscopic scale of hundreds of microns (see Supplementary Fig. 9). A key result of this study is the fact that switching the polarization state with a single, in-plane pulse leads to the deterministic switching and formation of a single multiferroic domain (details in Supplementary Notes 3–5). However, for a field applied along the [010] direction, that is, Pt wires parallel to [100], a blocky multi-domain case persists even in the poled region (Supplementary Figs. 6–9). In this multi-domain case, upon poling, the

domains are locally switched (Supplementary Fig. 10) due to a local polarization reversal, where the domain wall boundaries (or antiphase boundaries, Supplementary Fig. 11) do not move. This asymmetric behavior can be attributed to the anisotropic strain from the substrate (Supplementary Note 1), preventing the formation of a macroscopic domain in the device [100]. In devices with electrodes parallel to [1̄10] and [110], a single ferroelectric domain is formed which can be expected since a component of the electric field points along [100], allowing the antiphase boundaries to nucleate and move with the field. We can now use such a single-domain multiferroic as a model system to understand the stability of the spin cycloid and the corresponding spin transport.

To probe the spin transport, first, an in-plane electric field was applied between the source and detector wires, as indicated in Fig. 2a. Following each electrical pulse, a low-frequency (7Hz) alternating current is introduced into the source wire, generating a magnon spin current through the spin Seebeck effect. Subsequently, a non-

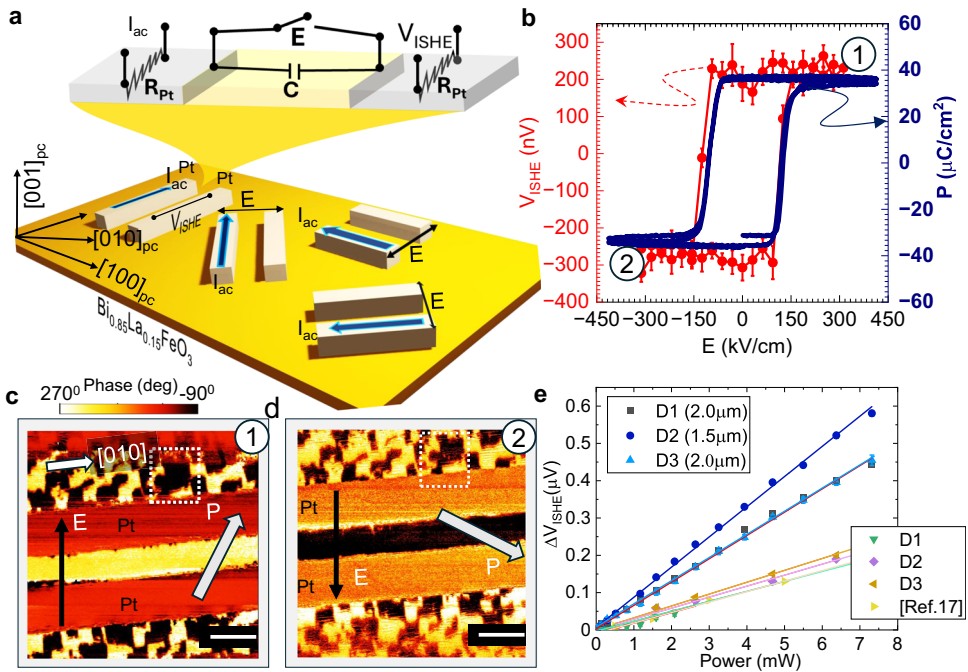

**Fig. 2 | Electric field control of magnons in Bi₀.₈₅La₀.₁₅FeO₃. a** Nonlocal magnon-transport measurement scheme in $Bi_{0.85}La_{0.15}FeO_3$ with Pt as a source/detector for spin-charge interconversion via magnon-transport. A resistive circuit schematic of in-plane devices where $R_{Pt}$ is the resistance of Pt electrodes, and $C$ is the capacitance of $Bi_{0.85}La_{0.15}FeO_3$, where the electric field is mainly distributed. The spin cycloid propagation vector $k$ is set by the $Bi_{0.85}La_{0.15}FeO_3$ polarization, which is controlled by an external in-plane electric field ($E$). The coordinate system uses pseudocubic notations. **b** Polarization and quasi-static magnon hysteresis as a function of external electric field. The blue line represents the polarization as measured by a Radiant Technologies ferroelectric test system (right axis) and the red circles correspond to the non-local ISHE voltage raw data (left axis). Error bars in ISHE voltage represent the standard statistical variation of lock-in voltages from the least-squares analysis measured over 150 s. **c, d** The corresponding PFM images after electrical poling in two opposite directions (labeled by labeled by `1' and `2' in (**b**)). PFM images were recorded in the same area, as marked by the rectangles. The scale bar is 2 μm. Bright and dark contrast arises from single ferroelectric domains between the Pt electrodes. Arrows represent the direction of the electric field `E' and corresponding polarization `P' in a single domain. **e** Differential voltage ($\Delta V_{ISHE}$) recorded in [010] devices as a function of the power injected into the source. Each data point is presented after averaging out to 150 s. The $Bi_{0.85}La_{0.15}FeO_3$ data presented were recorded in several devices (represented as D) with the same orientation and compared with the non-local voltage data belonging to the BiFeO₃ (100 nm)/Pt(6 nm) with the spacing of 2 μm. The data is reproduced from ref. 17 and corresponds to a metal electrode spacing of 1 μm. In the case of BiFeO₃, the ferroelectric domains were stripes whereas $Bi_{0.85}La_{0.15}FeO_3$ data was recorded in a ferroelectric single domain state. Lines are linear fit to the data.

equilibrium magnon spin accumulation at the $Bi_{0.85}La_{0.15}FeO_3$ interface underneath the Pt detector initiates the flow of spin angular momentum into the adjacent Pt. The resulting spin current is then converted into a measurable voltage through the Inverse Spin Hall Effect (ISHE) of Pt (Supplementary Note 6), and the signal is lock-in detected at $2\omega$. Each data point is averaged over a duration of 150 s. The ferroelectric polarization hysteresis was measured at 5 kHz ("Methods"), and the corresponding ISHE hysteresis was recorded in a remnant state where an electric field was applied only to set the polarization state and removed during the nonlocal voltage ($V_{ISHE}$) measurement. The nonlocal voltage hysteresis precisely reflects the ferroelectric polarization response (Fig. 2b, red data), indicating the existence of polarization-controlled magnon transport. Notably, in the [010] devices, the electric field and therefore, the polarization $P$ has the capacity to control the sign of the magnon spin current flowing through the $Bi_{0.85}La_{0.15}FeO_3$. This nonvolatile electric field magnon switching is illustrated in Supplementary Note 7, Fig. 27, where the ferroelectric polarization deterministically controls non-reciprocal magnon transport in the $Bi_{0.85}La_{0.15}FeO_3$.

Similar experiments on BiFeO₃ with a stripe domain structure were performed and a comparison is presented in Fig. 2e. The data corresponding to BiFeO₃ is also reproduced from Parsonnet et al.[17] The data from the different (D) devices corresponds to the 71° BiFeO₃ and reported data from Parsonnet et al.[17] belongs to the 109° BiFeO₃. We find that the $Bi_{0.85}La_{0.15}FeO_3$ has a consistently higher voltage output than the BiFeO₃ (by ~ 400% at the equivalent spacing). Furthermore, we find that the magnitude of the electric field required to switch the magnon

spin current is indeed significantly smaller (Supplementary Fig. 26), consistent with prior studies[19]. This doubly confirms the key advantages of single-domain $Bi_{0.85}La_{0.15}FeO_3$ over its parent compound.

The strong enhancement in the inverse spin Hall voltage for the $Bi_{0.85}La_{0.15}FeO_3$ compared to BiFeO₃ prompts us to explore the microscopic differences, if any, in the magnetic structure, particularly the spin cycloid. We used a combination of imaging techniques (PFM and second harmonic generation(SHG)-linear dichroism to probe the ferroelectric state and NV magnetometry to probe the spin cycloid, details in "Methods"). A comparison of the ferroelectric domain structure and corresponding magnetic (spin cycloid) is presented in Fig. 3. To determine the local directions of the polarization in each domain (discussed in Fig. 2c, d), optical SHG is used to map the ferroelectric domains in oppositely poled states (Fig. 3a, b). The red and blue areas correspond to domains with orthogonal in-plane polarization, and it is clear that in the device [010], the in-plane polarization is switched by 90° upon poling with oppositely directed fields. NV microscopy (Fig. 3c, d) reveals the presence of uniform spin cycloids in the oppositely poled domain. It is noteworthy that the sense of the cycloid stripes has rotated by 90 degrees, between these two switched states. This observation reveals that the ferroelectric single domains prefer to form a single variant cycloid, consistent with previous results[26]. We can conclude that the polarization is parallel to the spin cycloid stripes, which leads us to conclude that $P$ is orthogonal to the propagation vector $k$ (drawn schematically in Fig. 3e, f), a result that is consistent with previous works[18,26,38]. It is also validated by poling the $Bi_{0.85}La_{0.15}FeO_3$ devices at different angles with respect to the direction

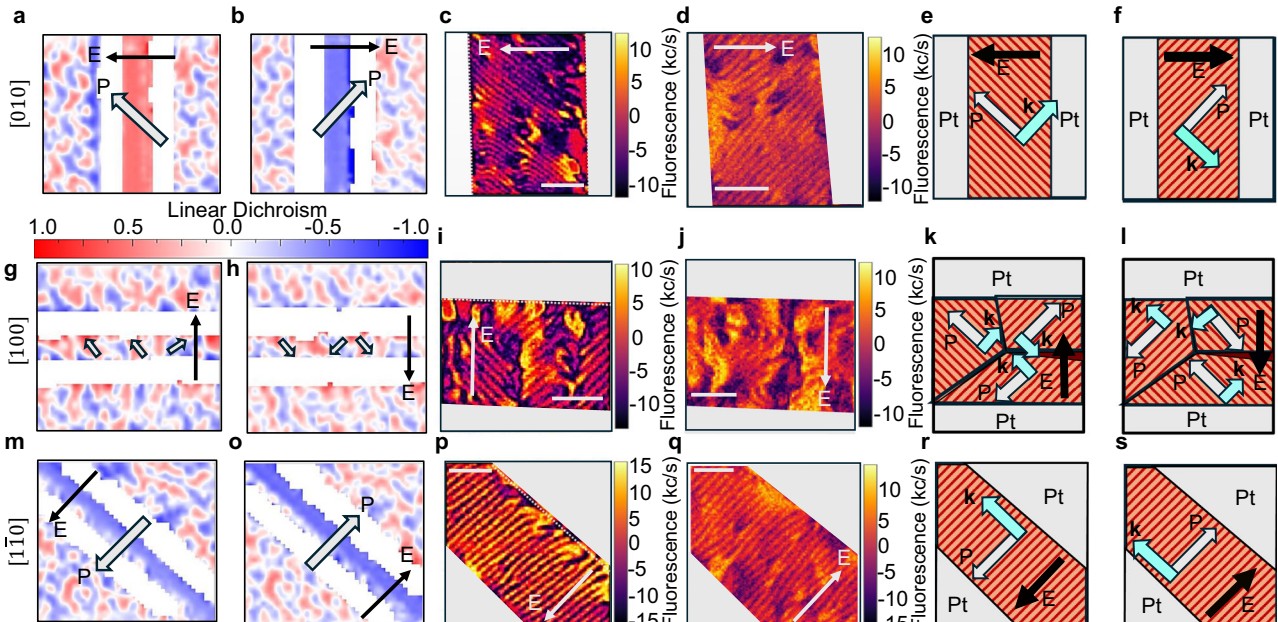

**Fig. 3 | Electric field control of magnonics based on the controllable magnetic and polar texture. a**–**f**, **g**–**l**, and **m**–**s** SHG-linear dichroism polar maps, iso-B NV images, and corresponding schematic for devices [010], [100] and [1̄10]. SHG (for in-plane polarization) and iso-B NV (for spin magnetic texture) are recorded for two opposite poling directions. Dark arrows represent the direction of the electric field and gray arrows show the polarization direction in specific domains. The gray (Pt) pads are used for the application of an in-plane electric field. The solid arrows indicate the direction of the applied electric field. The stripe patterns in NV images

are indicative of the canonical spin cycloid as observed in previous studies[18, 26, 38]. The line scans on the iso-B NV images (Supplementary Information Fig. 19) show that the period does not change appreciably after electric field switching in a single domain region. The opposite poled NV images were recorded on different devices. Schematics highlight the relationship between the polarization $P$ and spin cycloid propagation direction $k$ deduced from NV measurements (Also see Supplementary Information Fig. 20). The scale bar in NV images is 500 nm.

of the spin cycloid and the ferroelectric polarization. The multi-domain device [100] has two variants of cycloid corresponding to the two ferroelectric domains (Fig. 3g–l), whereas the same contrast in [1̄10] (in opposite poling) indicates 180° switching (Fig. 3m–s). Despite the same $k$ cycloid in 180° switch, $P$ switching will change the hand-edness in the opposite poled state[39]. With these insights, we conclude that the magnetic and ferroelectric order parameters are intimately tied in $Bi_{0.85}La_{0.15}FeO_3$ in a similar fashion to $BiFeO_3$[26, 38], and we show how the polarization and cycloid behave under electric fields pointing in different directions.

To probe the effect of such a single variant spin cycloid in the single-domain ferroelectric state, we proceeded to measure the non-local spin transport through the same test structures described in Fig. 3a–s, electric field dependent inverse spin Hall voltage hystereses were measured along these crystallographic directions under the same protocol as discussed in Fig. 2a, b. The single-domain devices [010] show ISHE voltage hysteresis (in BLACK) that corresponds to their ferroelectric hysteresis (Fig. 2b). Strikingly however, the multi-domain [100] oriented device does not show any appreciable ISHE hysteresis (in GREEN) despite exhibiting a clear ferroelectric hysteresis (Supplementary Fig. 8). Insight into this is readily obtained from the NV magnetometry images shown in Fig. 3i, j which shows no change in the topography of the spin cycloid; this is also schematically captured in Fig. 3k, l. Thus a balanced signal between the two types of domains with opposite contributions is likely to show a net null signal in the case of device [100]. This reveals that not only is magneto-electric coupling important but also the uniform magnetic *cycloid* is required for effective magnon spin flow. The behavior exhibited by the [010] device serves as a key to understanding the [100] device's behavior. A spin cycloid propagation vector of $k = [1\bar{1}0]$ results in a positive ISHE signal and $k = [\bar{1}10]$ yields a negative ISHE signal, as shown in Fig. 4a; it follows that a combination of domains with $k = [\bar{1}10]$ and $k = [1\bar{1}0]$, as observed in the [100] device, leads to a null signal without any discernible

magnon spin hysteresis. Although the precise correlation between the direction of $k$ and the spin carried by a magnon current would be interesting, the present observations affirm that the direction of $k$ holds greater significance than the net polarization in determining the non-local magnon signal.

Within the [1̄10] device, illustrated in Fig. 4, we note that the [1̄10] ([110]) devices have a lower magnitude with a positive (or negative) offset. The sign of the offset is consistent from device to device (5 devices for each orientation), as discussed in Supplementary Note 7, Fig. 29. A finite magnon output in the two devices where the single-domain ferroelectric with one variant spin cycloid is present can be understood from the symmetry of $P$ and $k$, if we consider that La-substitution can allow for different symmetry operations when switching the polarization[18, 38]. With an electric field along [110], in the parent $BiFeO_3$, this would result in a $C_2$ rotation about [001], or two successive 71° switches within the (001) Fig. 4b. Moreover, with electrical poling, the ferroelastic domains (71° or 109°) in $BiFeO_3$ remain the same when the net polarization and electric field are parallel, whereas the domains re-orient themselves if the electric field is orthogonal to the polarization in the plane devices[38]. This leads to the magnon transport in all directions of approximately the same magnitude. Furthermore, the magnon transport anisotropy in the spin cycloid is evident where magnons propagating along $k$ give larger output than its orthogonal direction[18] in the case of $BiFeO_3$ having 109°, whereas in $BiFeO_3$ with 71° does not show any anisotropy due to its zig-zag spin cycloid pattern. In the case of La-substituted $BiFeO_3$, the ferroelectric domain becomes a single domain after electrical poling (the unpoled pristine state does not favor magnon transport, as noted in Supplementary Information Note 7, Fig. 27), providing a key degree of freedom to choose a single variant, free from ferroelastic domain wall scattering in the magnon transport. As noted above in the case of $BiFeO_3$, $k$ is rotated about the [001], to which it is orthogonal, resulting in $k \rightarrow -k$. In the case of La-substituted $BiFeO_3$, however, the polarization along [112] may allow for this rotation

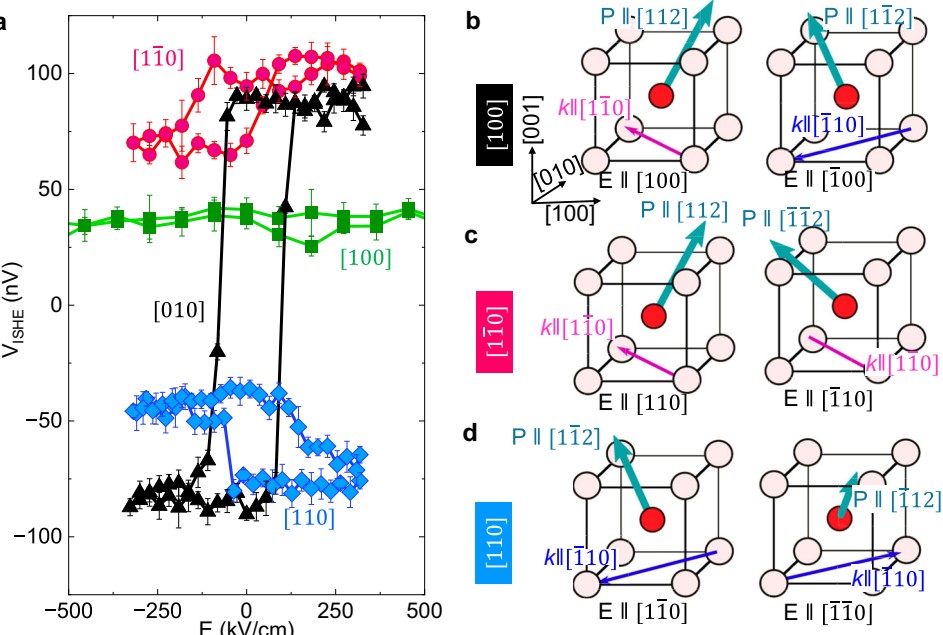

**Fig. 4 | Anisotropic magnon transport. a** Magnon-generated nonlocal ISHE voltage hysteresis was measured as a function of the external in-plane electric field in devices with four different orientations labeled by the pseudocubic direction with respect to the orientation of the Pt wires. The power in the source electrode was fixed to 2 mW ($I_{ac}$ = 1.7 mA). A depiction of the response of the $Bi_{0.85}La_{0.15}FeO_3$ unit cell to different poling fields is given in (**b**–**d**) for device orientations [010], [$\bar{1}$10], and [110] respectively. $k, E, P_{net}$ represent the propagation vector of the spin cycloid, in-plane electric field, and net in-plane ferroelectric polarization. The relation between $P$ and $k$ is drawn based on the experiment performed in Fig. 3, where $P$ and $k$ are mapped out, and are consistent with prior studies of the cycloid

in $BiFeO_3$[18,26,38]. In (**b**), the direction of $P$ is set to [112] in a single domain after poling where [$\bar{1}$10] is the allowed spin cycloid direction (see Supplementary Note 6, Fig. 19). In the opposite poling configuration, the direction $P$ is switched to [1$\bar{1}$2] (see also Fig. 3(a–d)) where the corresponding spin cycloid propagation vector will be [$\bar{1}$10]. Similarly, a single domain device $\bar{1}$10 or [110] switched 180° in the in-plane (c.f. Fig. 3(m–o)) in opposite poling since the orientation of Pt electrodes decides the direction of $P$ and underneath $k$. The schematics are only considered here for single-domain devices and for the multi-domain device [100] where the ISHE voltage change as a function of the electric field is found to be negligible (green data in (**a**)), more explanation is given in the Supplementary Note 6, Fig. 20.

to happen about the [1$\bar{1}$0] or [$\bar{1}$10] direction, rather than about the film normal (Fig. 4c, d). This operation, for example from [112] to [1$\bar{1}$2] is only a ~70° rotation of $P$, rather than the two 71° events to rotate around [001]. In addition, the rotation axis in this scheme is parallel to $k$, which then does not change sense after the operation. We would expect this to result in a small magnon signal, as observed. From the controlled experiments of the magnon transport through multi-domain versus single-domain state (Supplementary Note 7, Fig. 27), it is suggested that the uniform spin cycloid is responsible for finite magnon output in a single-domain multiferroic. Furthermore, the orientation of the spin cycloid's propagation vector decides the inverse spin Hall voltage output's magnitude. This indicates that the anisotropic nature of magnon transport as is intricately linked to the spin cycloid and, thus, the polarization of the $Bi_{0.85}La_{0.15}FeO_3$.

In summary, our study demonstrates the effective transmission of magnons in lanthanum-substituted $BiFeO_3$, resulting in a multiferroic material that can be polarized into a stable, non-volatile, uniform ferroelectric domain with a single variant of the spin cycloid. This stands in contrast to pure $BiFeO_3$, where the coexistence of two variants in both spin cycloids and stripe-like ferroelectric domains leads to a diminishing magnon signal. We observe that – by suitably choosing the direction of the applied electric field – it is possible to maximize or cancel the effect of ferroelectric switching on magnon transport. This research provides a means to customize ferroelectric domains and complex antiferromagnetic spin cycloids, as well as to understand the resulting spin transport, offering a pathway to design the single domain multiferroics for efficient magnon transport for future applications in magneto-electric spin-orbit logic and memory.

## Methods

### Thin film deposition

$BiFeO_3$ and Lanthanum (La) substituted $BiFeO_3$ ($Bi_{0.85}La_{0.15}FeO_3$) thin films were prepared by pulsed laser deposition (PLD) in an on-axis geometry with a target-to-substrate distance of ~50 mm using a KrF excimer laser (wavelength 248 nm, COMPex-Pro, Coherent) on $DyScO_3$(110)substrates. $DyScO_3$ substrate has a close lattice match (-0.3%) that helps in the high-quality epitaxial $BiFeO_3$ thin film growth. Film thickness was fixed to 90 nm unless otherwise specified. Before the deposition, the substrates were cleaned with IPA and Acetone for 5 min each. The substrates were attached to a heater using silver paint for good thermal contact. $BiFeO_3$ and $Bi_{0.85}La_{0.15}FeO_3$ layers were deposited with a laser fluence of 1.8 $Jcm^{-2}$ under a dynamic oxygen pressure of 140 mTorr at 710 °C with a 15 Hz laser pulse repetition rate. The samples were cooled down to room temperature at 30 °C/min at a static $O_2$ atmospheric pressure. The prepared samples were immediately transferred to a high vacuum DC magnetron sputtering chamber for Pt deposition. 15 nm of Pt was sputtered at 15 W power at room temperature in a 7 mTorr dynamic Ar atmosphere. The thicknesses were calibrated using X-ray reflectivity and atomic force microscopy.

### Crystal structure determination

The crystal structures of both $BiFeO_3$ and La-substituted $BiFeO_3$ were determined through X-ray diffraction, utilizing a high-resolution X-ray diffractometer (PANalytical, X'Pert MRD). The symmetric line scan ($\theta$-$2\theta$) employed a fixed-incident-optics slit set at 1/2°, while the reciprocal space mapping (RSM) involved an asymmetric 2D scan with a slit of 1/32°. The X-ray source used the Cu K$\alpha$ transition

(wavelength: 1.5401 Å), and detection employed a PIXcel$^{3D}$-Medipix$^3$ detector with a fixed receiving slit of 0.275 mm. Also see Supplementary Note 1.

### Cross-section sample preparation and high-angle annular dark field scanning transmission electron microscopy (HAADF-STEM)

The cross-section samples were prepared using a Helios660 scanning electron microscope/focused ion beam (SEM/FIB) with a gallium (Ga) ion beam source. After sample preparation, the cross-section samples were analyzed using an FEI Titan Themis G3 scanning transmission electron microscope (STEM) equipped with double correctors and a monochromator. High-angle annular dark-field scanning transmission electron microscopy (HAADF-STEM) imaging was performed at 300 kV accelerating voltage. Fourier-filtered HAADF-STEM images were analyzed using CalAtom software to extract the atomic position of Bi/La and Fe ions by multiple-ellipse fitting. The Fe displacement vector in each unit cell was calculated by confirming the center of mass of its four closest Bi/La neighbors. The displacement vector D of the Fe column is represented as follows:

$$\mathbf{D} = \mathbf{r}_{Fe} - \frac{\mathbf{r}_1 + \mathbf{r}_2 + \mathbf{r}_3 + \mathbf{r}_4}{4}, \tag{1}$$

where $\mathbf{r}_{Fe}$ is the position vector of the Fe column. $\mathbf{r}_1, \mathbf{r}_2, \mathbf{r}_3, \mathbf{r}_4$ are the position vectors of the four closest Bi/La neighbors in each unit cell. The color of the displacement vectors was represented by the vector magnitude. The visualization of the two-dimensional atomic displacement was carried out using Python. Calculation of the net displacement according to the unit cell projection is discussed in Supplementary Note 2.

### Ferroelectric domain characteristics

Piezoresponse force microscopy (PFM) imaging was conducted employing the MFP-3D system from Asylum Research, featuring Dual AC Resonance Tracking (DART) mode. Throughout the imaging process, the system operated in lateral mode, ensuring accurate lateral resolution in the acquired images. For these measurements, a silicon cantilever coated with platinum (Pt) was utilized, serving as a conducting electrode for the precise and localized application of an electric field. See Supplementary Note 3 for further information.

### Optical second harmonic generation for in-plane polarization mapping (SHG)

These measurements were conducted in a normal-incidence reflection geometry on poled devices. Light excitation was achieved using a Ti/sapphire oscillator with ∼100 fs pulses, a center wavelength of 900 nm, and a 78 MHz repetition rate. To manipulate the incoming light's polarization, a Glan-Thompson polarizer was employed, followed by passage through a half-wave plate. The polarized light then traversed a short-pass dichroic mirror and was focused onto the sample using a 100x objective lens with a numerical aperture (NA) of 0.95. The back-reflected SHG signal passed through a short-pass filter and was detected using a spectrometer (SpectraPro 500i, Princeton Instruments) equipped with a charge-coupled device camera (Peltier-cooled CCD, ProEM+:1600 eXcelon$^3$, Princeton Instruments). Diffraction-limited confocal scanning microscopy was employed to generate SHG intensity maps. At the sample location, a commercial Thorlabs polarimeter verified the incoming light's polarization incident on the sample and the light polarization entering the detector. Linear dichroism maps were constructed through the subtraction of SHG intensity maps with incident light polarization along [110]$_{pc}$ or [1$\bar{1}$0]$_{pc}$ directions. The poling process was performed ex-situ for all devices. See Supplementary Note 4 for further information.

### Scanning nitrogen-vacancy (NV) microscopy

The magnetic texture in the samples was imaged at room temperature utilizing a commercial scanning NV magnetometer (Qnami ProteusQ). Scanning NV magnetometry has been described extensively elsewhere; briefly, a parabolically-tapered diamond cantilever (Quantilevel MX+) was used to detect the stray fields from the sample by probing the frequency shift of the NV center spin as the tip was scanned across the surface. To facilitate wide-area scans, data was collected in the "iso-B" mode, where the peak shift is estimated from the microwave response at two frequencies rather than the full spectrum (e.g., ref. [40]). Iso-B measurements were validated against select measurements of the full spectrum to ensure the magnetic texture is reported faithfully (See Supplementary Note 5).

### Device fabrication

The sample fabrication started with sonication in acetone and isopropyl alcohol. Subsequently, a positive photoresist (MIR 701), ∼500 nm thick, was uniformly coated at 7000 RPM for 60 s using a spin coater. The coated sample was then baked at 100 °C for 60 seconds. Photolithography was executed through a Karl Suss MA6 Mask Aligner, with i-line exposure at 10 mW/cm$^2$ for 5 s. Following exposure, the resist underwent wet-etching using MEGAPOSIT MF-26A photoresist developer for 20 s. Subsequently, the Pt layer was ion-milled down to the multiferroic film surface (Intlvac Nanoquest, with a Hiden Analytical SIMS), resulting in the formation of rectangular stripes measuring 120 μm × 1.3 μm. This process was conducted at the Marvell Nanofabrication laboratory at UC Berkeley.

### Spin transport measurements

Transport measurements were conducted employing 4-terminal devices, wherein two terminals were dedicated to source current injection, and the remaining two served as output terminals for inverse spin Hall effect (ISHE) voltage measurement. One source terminal and one detection terminal were also used to apply an electric field for ferroelectric polarization control. The entire experimental setup and procedures were orchestrated using an in-house developed Python code and a Keithley 7001 switch box, maximizing repeatability. To measure the nonlocal ISHE voltage ($V_{ISHE}$), an SR830 lock-in amplifier was synchronized to the second harmonic of the 7 Hz source current, isolating responses to the thermal gradients. This comprehensive setup allowed us to perform accurate and controlled transport measurements (using all automated codes), facilitating the investigation of electric field-controlled nonlocal voltage measurements. See Supplementary Note 7 for detailed information.

### Computational methods (effective Hamiltonian)

In the case of BiFeO$_3$, the magnetic ground state is a G-type antiferromagnetic configuration, which is modulated by the complex magnetic arrangement called a spin cycloid. The BiFeO$_3$ doped with rare-earth leads to further modulation in the magnetic texture or relaxed into a G-type configuration without the cycloid. To understand this complex state in BiFeO$_3$ and doped BiFeO$_3$ compounds, we performed Monte Carlo simulations governed by the first principle-based effective Hamiltonian. This effective Hamiltonian is expressed as follows for BiFeO$_3$ and doped BiFeO$_3$:

$$\begin{aligned} E_{\text{total}} &= E_{\text{FE-AFD}}(\{\mathbf{u}_i\},\{\omega_i\},\{\eta_H\},\{\upsilon_i\}) \\ &\quad + E_{\text{mag}}(\{\mathbf{m}_i\},\{\mathbf{u}_i\},\{\omega_i\},\{\eta_H\}), \end{aligned} \tag{2}$$

where the first term in equation (2) $E_{\text{FE-AFD}}$ (FE: ferroelectric, AFD: antiferrodistortion octahedral tilts) contains energy terms arising from the nonmagnetic variables (local mode ($\mathbf{u}_i$) being the parameter corresponding to the electric dipole (or the electrical polarization), global homogeneous ($\eta_H$) and Fe-centered inhomogeneous strain tensor ($\upsilon_i$). $\omega_i$ is the oxygen octahedral tilt representing the axis of rotation) and

**Table 1 | Calculated energy (ΔE) of different magnetic states in BiFeO$_3$ and Bi$_{0.85}$La$_{0.15}$FeO$_3$**

| Magnetic order | Bi$_{0.85}$La$_{0.15}$FeO$_3$ ΔE (meV/unit cell) | BiFeO$_3$ ΔE (meV/unit cell) |
|---|---|---|
| R3c cycloid | 0 | 0 |
| M1 F-AFM | 9.46 | 17.04 |
| M1 cycloid | 11.55 | 21.55 |
| M2 cycloid | 13.22 | 26.91 |

their coupling. The second term represents the magnetic mode of the BiFeO$_3$ ($m_i$ represents the magnetic moment at site $i$ centered at the Fe ion with its magnitude fixed ($4\mu_B$)) and its coupling with other modes. The expansion of this term is as follows:

$$\begin{aligned}
E_{mag}(\{\mathbf{m}_i\},\{u_i\},\{\omega_i\},\{\eta_i\}) = &\sum_{i,j,\alpha,\gamma} Q_{ij\alpha\gamma} m_{i\alpha} m_{j\gamma} + \sum_{i,j,\alpha,\gamma} D_{ij\alpha\gamma} m_{i\alpha} m_{j\gamma} \\
&+ \sum_{i,j,\alpha,\gamma,\nu,\delta} E_{ij\alpha\gamma\nu\delta} m_{i\alpha} m_{j\gamma} u_{i\nu} u_{i\delta} \\
&+ \sum_{i,j,\alpha,\gamma,\nu,\delta} F_{ij\alpha\gamma\nu\delta} m_{i\alpha} m_{j\gamma} \omega_{i\nu} \omega_{i\delta} \\
&+ \sum_{i,j,l,\alpha,\gamma} G_{ij\alpha\gamma} \eta_l(i) m_{i\alpha} m_{j\gamma} \\
&+ \sum_{i,j} K_{ij}(\omega_i - \omega_j) \cdot (\mathbf{m}_i \times \mathbf{m}_j) \\
&+ \sum_{i,j} C_{ij}(\mathbf{u}_i \times \hat{e}_{i,j}) \cdot (\mathbf{m}_i \times \mathbf{m}_j).
\end{aligned} \tag{3}$$

Here the I$^{st}$ term represents the magnetic dipolar interaction. The II$^{nd}$ term corresponds to the magnetic exchange coupling up to the third nearest neighbor. The III$^{rd}$, IV$^{th}$, and V$^{th}$ terms describe the change in the magnetic exchange interaction induced by the local polar mode, AFD tilt, and strain. An important point to note is that the first five energy terms lead to the collinear magnetism in BiFeO$_3$. The VI$^{th}$ term involving octahedral or AFD tilting represents the Dzyaloshinskii-Moriya interaction (DMI) and is responsible for the weak magnetization in the AFM state of BiFeO$_3$. The last term of Eq. (4) is responsible for the cycloid (via the inverse spin-current effect which is a DMI effect), and it is the only term related to electric polarization. This energy allows the stable spin cycloid with $k$ being the propagation vector along [1$\bar{1}$0] (within (111) plane) (with P ∥ [111]) in BiFeO$_3$. All the coupling coefficients were calculated using Density Functional Theory for both pure BiFeO$_3$ as well as lanthanum-doped BiFeO$_3$. All the calculations were done for bulk stress-free supercells of $12 \times 12 \times 12$ unit-cells, both for pure and doped BiFeO$_3$. The complex modulated phases $M1$, $M2$ are phases found as a result of temperature cooling of rare-earth-doped BiFeO$_3$, further relaxed for (15%) La-substituted BiFeO$_3$ and represent modulated polar arrangements of periods of 6 and 4 unit cells, respectively. The calculated ground state energy values of the different magnetic textures such as pure cycloid and complex magnetic texture (mixed state of the cycloid and G-type antiferromagnet) are given in Table 1.

## Reporting summary

Further information on research design is available in the Nature Portfolio Reporting Summary linked to this article.

## Data availability

The data that support the findings of this study are available from the corresponding author upon reasonable request.

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

## Acknowledgements

We thank Sasikanth Manipatruni, and Dmitri E. Nikonov for fruitful discussions. This work was primarily supported by the U.S. Department of Energy, Office of Science, Office of Basic Energy Sciences, Materials Sciences and Engineering Division under Contract No. DE-AC02-05-CH11231 (Codesign of Ultra-Low-Voltage Beyond CMOS Microelectronics) for the development of materials for low-power microelectronics. I.H., H.Z., and R.R. acknowledge the Air Force Office of Scientific Research 2D Materials and Devices Research program through Clarkson Aerospace Corp under Grant No. FA9550-21-1-0460. P.K. acknowledges support from the Intel Corporation as part of the COFEEE program. S.Z. and L.C. acknowledge funding from the Brown School of Engineering and the Office of the Provost. P.S. acknowledges support from the Massachusetts Technology Collaborative, Award number 22032. L.W.M. and R.R. also acknowledge partial support from the Army/ARL as part of the Collaborative for Hierarchical Agile and Responsive Materials (CHARM) under cooperative agreement W911NF-19-2-0119. X.L. and Y.H. are supported by the Welch Foundation (C-2065-20210327). Y. H. acknowledges the support from NSF-2329111 and NSF-2239545). We acknowledge the Electron Microscopy Center, Rice. J.I.-G. acknowledges support from the Luxembourg National Research Fund through grant C21/MS/15799044/FERRODYNAMICS. M.R. and D.S. acknowledges the Army Research Office under the ETHOS MURI via cooperative agreement W911NF-21-2-0162. B.X. acknowledge financial support from National Natural Science Foundation of China (Grant No. 12074277) and Projects of International Cooperation and Exchanges NSFC (Grant No. 12311530693) S.M. and L.B. would like to acknowledge the ARO Grant No. W911NF-21-1-0113, the MURI ETHOS Grant No. W911NF-21-2-0162 from the Army Research Office (ARO), the U.S. Department of Defense under the DEPSCoR program (Award No. FA9550-23-1-0500) and the Vannevar-Bush Faculty Fellowship (VBFF, Grant No. N00014-20-1-2834). The calculations were performed at the Arkansas High Performance Computing Center (AHPCC).

## Author contributions

R.R. and S.H. conceived the idea and designed the experiments. S.H. and I.H. performed thin film deposition, measurements, and analysis. I.H. patterned the devices and wrote the experimental procedure scripts with the inputs from S.H. P.M. and P.S. performed the Nitrogen-Vacancy (NV) magnetometry. S.M. performed theoretical calculations under the supervision of B.X., J.I.-G., and L.B. X.L. performed cross-sectional microscopy and polarization mapping under the guidance of Y.H. M.R. did controlled sample preparation under the supervision of D.S. M.R and S.Z. performed some NV measurements with the help of L.C. and P.S. P.B. performed second harmonic generation (SHG) mapping. H.T. performed controlled sample lithography under the supervision of J.G.A. J.K., P.K., T.Y.K., and H.Z. gave suggestions on the PFM experiments. S.S. and L.W.M. gave suggestions and commented on the manuscript. R.R. and Z.Y. supervised the work. S.H. and I.H. wrote the manuscript. All authors have participated in the discussion and reviewed the manuscript.

## Competing interests

The authors declare no competing interests.
