## [Peer Review File · Nature Communications]

Non-volatile magnon transport in a single domain multiferroicREVIEWER COMMENTS

Reviewer #1 (Remarks to the Author):

Sajid Husain et al. report the transport of thermal-excited magnon current in single domain La doped BiFeO₃ and shows the non-volatile control of second harmonic signal by reversing the in-plane ferroelectric polarization component. The single domain BiFeO₃ is achieved by La doping BiFeO₃ and then applying voltage to induce phase transition. They observe that the non-volatile electric field control of second harmonic signal shows strong in-plane anisotropy, which is demonstrated to be connected to the coupling between ferroelectric polarization P and spin cycloid direction K. I find the results are interesting to the spintronic society. However, the authors need to clarify the following concerns.

1. The title 'Non-volatile spin transport in a single domain multiferroic'. To the best of my knowledge, the spin transport signal should be detected by first harmonic signal (Nature Physics 11, 1022-1026 (2015). Nature Nanotechnology, 15, 563-568 (2020).). The second Harmonic signal mainly detect the thermal effect, spin Seebeck effect that correlated to thermal magnon current (Nature Physics 11, 1022-1026 (2015).; Physical Review Letters, 129, 087601 (2022).). In the article, the authors only show the second harmonic signal. I suggest the author to clarify the concept and change the title to avoid misunderstanding.
2. Can the authors plot the first harmonic signal? Is there any non-volatile anisotropic control? This should be directly connected to 'spin transport'.
3. Can the author use magnetic field to control magnetic moments in La:BFO to help demonstrate the magnetic origin of the second Harmonic signal?
4. In the similar BFO/DSO heterostructures with two-variant domain structure, can the authors observe the anisotropic non-volatile electric field control of second harmonic signal? I think this is an important control sample that helpful to the discussion on symmetry in page 7.
5. There are some typos should be corrected. For example, the Extended data title is inconsistent with the main text. The notation of electric field E (as well as K) in Figure 4c and Figure 4d seem not consistent with each other (left to right panel). The authors should revise the manuscript carefully.

Reviewer #2 (Remarks to the Author):

This paper by Husain et al. is focused to understand the formation of single domain multiferroic as a model system for efficient spin transport. In the process Bi_{0.85}La_{0.15}FeO₃ has been deposited DyScO₃ substrate. While the paper is more focused to the fabrication and characterization of the material, the physics to understand the mechanism behind is not clearly explained. The experimental section is technically sound and the observations are well explained. However, the paper lacks novelty considering the fact the effect of La substitution in BiFeO₃ is already reported in the literature. Going through the paper, I have certain observations furnished below.

1. In the PFM images why the in-plane polling leads to the formation of a single ferroelectric domain? This has to be discussed clearly.
2. The authors have to mention clearly the novelty of this work in terms of concluded inferences in contrast to the papers reported in the literature.

Minor Comments:

The abbreviated nomenclatures such as DMI , SHG, P may be clearly mentioned in the text.

The decision on the acceptance of this paper rests solely on the novelty of the work albeit the experimental results and there explanations are scientifically sound.

The paper, presented by Husain et al., investigated the non-volatile spin transport phenomenon in single domain multiferroics BiFeO₃ and La – substituted BiFeO₃ compounds from experimental as well as theoretical studies. Their studies offer a new pathway to design model systems using the single-domain multiferroics for efficient spin magnon transport in future applications. The work is indeed interesting. However, the manuscript can be accepted for publication in Nature Communications if the authors well resolve the following issues to improve the current version of the manuscript.

Q # 1. Rhombohedral BiFeO₃ in its G-type antiferromagnetic state shows Rashba splitting which is intrinsically linked with spintronics. Author should mention this in the introduction section of the manuscript.

Q # 2. Since, BiFeO₃ exhibits non-centrosymmetric crystal structure, authors should draw an analogy between the non-centrosymmetry and multiferroic behaviour of the system.

Q # 3. Authors have considered only 15% La – substitution, why? Authors could use different La concentration and compare their results accordingly in this work.

Q # 4. Why does the direction of the spontaneous polarization alter upon La doping on BiFeO₃?

Q # 5. Authors have studied the inverse piezoelectric effect which is accompanied by the relationship between mechanical deformation in materials subjected to an applied electric field. Can authors comment on the amount of pressure or strain that has been formed inside the compounds?

Q # 6. Why in-plane poling leads to the formation of a single ferroelectric domain and why this is the novel feature for La-substituted BiFeO₃?

Q # 7. In Figure 2, it will be Bi_{0.75} instead of Bi_{0.85}. Authors should correct it.

Q # 8. Authors have studied the inverse spin Hall voltage. Can authors comment on the estimation of spin Hall conductivity associated with the spin current?

Q # 9. The magnitude of electric field required to switch the magnon spin current is found to be significantly smaller for doped system in contrast to the parent compound. Why?

Q # 10. Authors should provide the effective values of second-order nonlinear susceptibility for both bare and doped systems through SHG method. Why did authors not implement the third harmonic generation for in-plane polarization mapping?

Q # 11. Why did authors use DyScO₃ substrates for film deposition?

Q # 12. Authors have considered the magnetic moment of Fe ion only in the Monte Carlo

simulations, why? As the La ion also exhibits some magnetic moment in the doped system.

Q # 13. Authors should provide some realistic names of future applications at the end of conclusion/ summary.

Q # 14. Authors should provide the full form of terms like DMI, SHG etc. when they appear first in the manuscript. Authors must carefully check the Supplementary Note numbers like 3 instead of 2 for “Ferroelectric Domain Characteristics”, 4 instead of 3 for “Optical second harmonic generation for in-plane polarization mapping (SHG)” etc. before resubmission.

Reviewer #3 (Remarks to the Author):

In this work, the authors report on the demonstration of control of magnon transport with electric field in La-substituted BFO. This material offers enhanced properties compared to regular BFO, as it can be prepared in a single ferroelectric domain state with in-plane electrodes and produces a stronger magnon signal. I found that the paper is rather well-written, one can follow the arguments without too much difficulty, and plenty of details are provided in the Methods and supplement. This is an interesting spin transport experiment, but in my opinion, this is mainly an improvement of what is reported in refs. 30, 31 and 35 (from the same authors) and is therefore not suitable for publication in Nature Communications but rather in a more specialized journal. I also regret that not much explication of the mechanism of the observed effect is provided and the reason why it is so much more efficient in this material is not really given.

More precisely, ref. 35 shows similar magnon transport experiments than here, but realized on regular BFO, so with stripy ferroelectric domains and a smaller non-reciprocity observed, while ref. 30 focuses on the switching of ferroelectric domains in regular BFO using electrodes and ref. 31 achieves also magnon transport between electrodes with a single cycloid wavevector in the domain in-between the electrodes by tuning the strain applied on the film with the substrate. In ref 31. similar results as here are obtained regarding the effect of the direction of k .

In addition, I have numerous remarks and questions about the article, which I provide below to help the authors improve their manuscript.

Figures

=====

There is in my opinion a general lack of rigor and clarity in the design of the figures (in main text and extended data). None of the STEM, SHG, PFM or NV maps has a colorscale specifying the displayed quantity and its values. This should be corrected. For the PFM, what is the probed polarization direction? For the NV data, one has to reach the very end and the Methods section to finally learn that this is iso-B data! This should be specified on the figures or in the captions!

Fig 1:

- Panel a: this plot is very strange to me, and I have the impression that it could actually be replaced by a table and become more readable. What are the dashed lines? Why use

different symbols for the different states (which are anyway placed at different positions along the x axis) rather than having one symbol for BFO and another one for BLFO? To which state on the graph does the "complex state" label refer?

- Panel b: I do not understand this figure. What is supposed to be varied on the x axis? Is it implied here that there would be 2 phases in BFO with different rotational sense of the cycloid? This rotational sense is fixed by the magnetoelectric DMI-term, so it is linked to P, which does not appear on the figure. In addition, the depiction of the "complex state" is not very clear. I think that this panel is not necessary.

- Panels c-d: I suggest to specify that the blue-green arrows represent P.

- Panel f: I am surprised by the curved wavefront of the cycloid on this image, usually BFO/DSO imaged with iso-B mode provides an undistorted zigzag pattern with straight lines and a 90° angle at the ferroelectric domain walls, which does not seem to be the case here. Can the authors comment on this? Is there thermal drift in their system despite the use of iso-B mode?

- Panel h: Why are the uniform looking patches designated as non-collinear phase in the image? The caption mentions a canted AFM phase, which seems right, but this is not a non-collinear phase.

Fig 2:

- Do not use P both for polarization and injected power.

- Along which direction is P in panel b?

- Panel e: What is the width of the gap between the electrodes for the BFO data? In the caption, domains are mentioned concerning BFO, I guess that the authors mean ferroelectric domains but this should be precised.

Fig. 3:

Why is the contrast so much weaker in the NV data in the column d,j,q? The cycloid period also looks different between both cases, especially in panels c and d, why? And I honestly cannot really see a cycloid in panel j, the data quality is not good enough.

Fig 4:

I find panels b, c, d a little difficult to understand.

Ext. Data Fig. 2:

In all cases except g-h, there all always two directions of P present between the electrodes, so I do not understand what the P arrows mean.

Ext. Data Fig. 4:

What is V_{nl} ?

Text

====

- Could the authors give more insight about why La-substitution changes the direction of P, and why including 15% of La decreases P by 50%? And why was 15% chosen? Is it actually an optimal value or chosen because it was experimentally convenient?

- End of 1st column, page 3: I do not understand. P is decreasing, thus reducing DMI, which is responsible for the stabilization of the cycloid and the tilt corresponding to the spin density wave. Why would this tilt increase when P decreases? Should it not decrease too? And if

there is actually an increase, it might be possible to detect it with scanning NV magnetometry. The field measured, which originates from the spin density wave, should be larger in BLFO in this case. Such data would be interesting.

Side note: the reference to the polar maps in Fig 1 here should be the panels c and d, rather than a and b.

- End of 2nd column, page 3: "a single, in-plane pulse leads to the deterministic switching and formation of a single multiferroic domain" : the magnetic state between the electrodes has not been mentioned in the text yet, so claiming that there is a single multiferroic domain sounds a little premature.

- 1st column, page 5: why is the signal so much larger (by a factor 4?) in BLFO compared to BFO? And if we compare with the experiment on TSO with a single k direction in ref 31, is it also so much larger?

- End of 1st column, page 6: a reference to Fig. 4 is missing.

- 2nd column, page 6: "topology of the spin cycloid": the authors should refrain from using the word topology here, this could be misleading as a cycloid is a topologically trivial state.

- 2nd column, page 6: "uniform magnetic texture": an antiferromagnetic cycloid with an attached spin density wave is a very complex non-collinear magnetic state, not a uniform state!

- 2nd column, page 6: I think that what would lead to a null magnon signal is a balanced combination of domains. Otherwise, if a domain with a given cycloidal state is dominant, would not the signal only be reduced?

- Beginning of 1st column, page 7: "This can be understood from the symmetry of P and k, if we consider that La-substitution can allow for different symmetry operations when switching the polarization." Can the authors be more specific here?

- 1st column, page 7: "This elucidates the anisotropic nature of magnon transport as it is intricately linked to the spin cycloid and thus the polarization of the BLFO": this is an overstatement. The anisotropic magnon transport is experimentally shown here, but not microscopically explained. And actually, it would have been extremely surprising if the magnetic cycloid had no effect on the spin waves.

Supplementary

=====

- p 26: in dual iso-B mode, 2 microwave frequencies are used, to get rid of the artifacts that could come from PL variation unrelated to the magnetic state.

- Suppl. Fig. 15: Why is the cycloid not visible in panel c? The bias field contribution should be subtracted from panel f.

Aurore Finco

Reviewer #4 (Remarks to the Author):

Response to comments from Reviewer #1

Remarks: Sajid Husain et al. report the transport of thermal-excited magnon current in single domain La doped BiFeO₃ and shows the non-volatile control of second harmonic signal by reversing the in-plane ferroelectric polarization component. The single domain BiFeO₃ is achieved by La doping BiFeO₃ and then applying voltage to induce phase transition. They observe that the non-volatile electric field control of second harmonic signal shows strong in-plane anisotropy, which is demonstrated to be connected to the coupling between ferroelectric polarization P and spin cycloid direction K . I find the results are interesting to the spintronic society. However, the authors need to clarify the following concerns.

Response: We thank the reviewer for finding our results interesting. We here provide a point-by-point response to suggestions/comments.

Comment#1. The title ‘Non-volatile spin transport in a single domain multiferroic’. To the best of my knowledge, the spin transport signal should be detected by first harmonic signal (Nature Physics 11, 1022-1026 (2015). Nature Nanotechnology, 15, 563-568 (2020).). The second Harmonic signal mainly detect the thermal effect, spin Seebeck effect that correlated to thermal magnon current (Nature Physics 11, 1022-1026 (2015).; Physical Review Letters, 129, 087601 (2022).). In the article, the authors only show the second harmonic signal. I suggest the author to clarify the concept and change the title to avoid misunderstanding.

Response: We thank the reviewer for the important comment. We agree with the reviewer on the part of the spin and thermal source of magnon generation in the first and second harmonics respectively. Nevertheless, the detection mechanism is the same owing to the inverse spin Hall effect despite the different origins of the magnon current. As per suggestions, we have changed the title to “**Non-volatile magnon transport in a single domain multiferroic**”. We once again thank the reviewer for this suggestion.

Comment#2. Can the authors plot the first harmonic signal? Is there any non-volatile anisotropic control? This should be directly connected to ‘spin transport’.

Response: We thank the reviewer for an important question. Due to the small spin Hall angle (0.03) of Pt, we did not observe any measurable first harmonic signal in our non-local devices, however, measurable with the large spin Hall angle material such as SrIrO₃ (SIO), and post-review as per the reviewer’s comment, we have prepared a new Bi_{0.85}La_{0.15}FeO₃/SIO sample to see the anisotropic behavior in the spin transport as shown in Figure R1 (below). We note, however, that the present work is focused more on the anisotropy of magnon transport in a single domain Bi_{0.85}La_{0.15}FeO₃, and we focus on understanding the physics of a single domain multiferroic on magnon transport compared to standard multidomain BiFeO₃ using Pt. If suggested, we can add this figure to the supplementary information.

Figure R1 : First harmonic data of the spin transport in Bi_{0.85}La_{0.15}FeO₃ (80nm)/SrIrO₃ (20nm) in the two device orientations.

Comment#3. Can the author use magnetic field to control magnetic moments in La:BFO to help demonstrate the magnetic origin of the second Harmonic signal?

Response: Thanks for the important comment. There are two ways we can use a magnetic field to understand the ferroelectric control; (i) the magnetic field should be larger than $\sim 16\text{T}$ [Nat Commun 6, 5878 (2015).] to release the spin cycloid; however, such a high magnetic field is beyond the reach of our PPMS. On the other hand, our focus is on the electric field to manipulate the magnetization. Furthermore, in a (ii) approach, even at smaller magnetic fields, the Nernst effect is dominant [supplementary Ref. 35] therefore it is not trivial to measure in a magnetic field. We have added the corresponding explanation in on page 3.

Comment#4 In the similar BFO/DSO heterostructures with two-variant domain structure, can the authors observe the anisotropic non-volatile electric field control of second harmonic signal? I think this is an important control sample that helpful to the discussion on symmetry in page 7.

Response: We thank the reviewer for this suggestion. We have added a discussion of the BiFeO_3 anisotropy on page 7 (ref.31).

Comment#4 There are some typos should be corrected. For example, the Extended data title is inconsistent with the main text. The notation of electric field E (as well as K) in Figure 4c and Figure 4d seem not consistent with each other (left to right panel). The authors should revise the manuscript carefully.

Response: We apologize for the typos; these are now corrected in the revised version. As per the Nature Communications guidelines, we have moved the extended figures to the supplementary information. We have checked the electric field and k vectors; they are consistent in both figures. The figure below is presented for the review process to show the E and k for both devices. If acceptable, we can add this to the supplementary information.

Figure R1: Electric field and spin cycloid propagation vector relations in two field directions.

Response to comments from Reviewer #2

Remarks: *This paper by Husain et al. is focused to understand the formation of single domain multiferroic as a model system for efficient spin transport. In the process Bi_{0.85}La_{0.15}FeO₃ has been deposited DyScO₃ substrate. While the paper is more focused to the fabrication and characterization of the material, the physics to understand the mechanism behind is not clearly explained. The experimental section is technically sound and the observations are well explained. However, the paper lacks novelty considering the fact the effect of La substitution in BiFeO₃ is already reported in the literature. Going through the paper, I have certain observations furnished below.*

Response: We thank the reviewer for finding our work sound and well-explained experiments and providing constructive comments to improve the manuscript. We agree that the La-substituted BiFeO₃ has already been reported, mainly in terms of the ferroelectric phase stability. We make two KEY points regarding WHY this present paper is distinctly different from what has been published on La substituted BiFeO₃ in the past. First, there has been no spin/magnon transport study in this system; substitution of La is known to change the spontaneous polarization magnitude and direction. As such, we expect that this will alter the DM vector as well. Second, the La substitution significantly changes the ferroelectric domain structure, as noted in the literature (Nat Commun 11, 2836 (2020)). This is important since this change in domain structure is critical to be able to electrically switch the La- BiFeO₃ and create a single ferroelectric domain structure. Thus, our current paper is NOT about La-BiFeO₃ but about spin/magnon transport in a single domain multiferroic. This then provides us with a pathway to study spin/magnon transport and its anisotropy without interference from scattering effects at the ferroelectric domain boundaries. In some sense, this is providing us with insights into the spin transport in a single domain multiferroic.

Comment#1. *In the PFM images why the in-plane polling leads to the formation of a single ferroelectric domain? This has to be discussed clearly.*

Response: We thank the reviewer for this comment. The multivariant to single variant switching in ferroelectric is well demonstrated in our previous work [Advanced Materials 35, 2301934 (2023)], where the local and global domain structure can be formed based on the direction of the net local polarization and the electric field with respect to the substrate axes. As discussed in Figure 3 and Supplementary Figure 7 and 8, a single domain can be formed based on the local (domain) polarization orientation with respect to the external electric field. Moreover, due to the La-substitution, it has more possible switching pathways than pure BiFeO₃ [Nat Commun 11, 2836 (2020)], which makes attaining a single domain possible. This effect is anisotropic due to the substrate strain anisotropy [Supplementary Note 1] consistent with the previous observations, Advanced Materials 35, 2301934 (2023). A relevant text is added in the revised supplementary and manuscript.

Comment#2. *The authors have to mention clearly the novelty of this work in terms of concluded inferences in contrast to the papers reported in the literature.*

Response: We make two KEY points regarding the novelty of our paper. First, there has been no spin/magnon transport study in the La-BiFeO₃ system. We note that the substitution of La is known to change the spontaneous polarization magnitude and direction. As such, we expect this will alter the DM vector as well. Second, the La substitution significantly changes the ferroelectric domain structure, as has been noted in the literature. This is important since this change in domain structure is important to be able to electrically switch the La-BiFeO₃ and create a single ferroelectric domain structure. Thus, the novelty of our paper rests on the study of spin/magnon transport in a single domain multiferroic, which we believe is fundamentally important to study. This then provides us with a pathway to study spin/transport and its anisotropy without interference from scattering effects at the ferroelectric domain boundaries. In some sense, this is providing us with insights into the spin transport in a single domain multiferroic. New text added in the introduction page-3:

Minor Comments:

The abbreviated nomenclatures such as DMI, SHG, and P may be clearly mentioned in the text.

Response: We thank the reviewer for the suggestions. The nomenclatures have been taken care of in the revised version.

The decision on the acceptance of this paper rests solely on the novelty of the work albeit the experimental results and there explanations are scientifically sound.

Response: We are grateful to the reviewer for the appreciation and suggestions. In our response letter as well as in the revised version of our paper, we have provided a detailed description of WHY this is a novel piece of work. We do hope that this is satisfactory. We are happy to elaborate some more if needed.

Response to comments from Reviewer #3

Remarks: *The paper, presented by Husain et al., investigated the non-volatile spin transport phenomenon in single domain multiferroics BiFeO₃ and La – substituted BiFeO₃ compounds from experimental as well as theoretical studies. Their studies offer a new pathway to design model systems using the single-domain multiferroics for efficient spin magnon transport in future applications. The work is indeed interesting. However, the manuscript can be accepted for publication in Nature Communications if the authors well resolve the following issues to improve the current version of the manuscript.*

Response: We are grateful to the reviewer for comments and suggestions to improve the manuscript with recommendation for publication. We provide a point-by-point response to comments/suggestions.

Q # 1. *Rhombohedral BiFeO₃ in its G-type antiferromagnetic state shows Rashba splitting which is intrinsically linked with spintronics. Author should mention this in the introduction section of the manuscript.*

Response: As per suggestions, we have added the discussion in the introduction.

Q # 2. *Since, BiFeO₃ exhibits non-centrosymmetric crystal structure, authors should draw an analogy between the non-centrosymmetry and multiferroic behaviour of the system.*

Response: As suggested, the discussion is added in the introduction.

Q # 3. *Authors have considered only 15% La – substitution, why? Authors could use different La concentration and compare their results accordingly in this work.*

Response: It is well established that the 15% La doping impacts the energy landscape significantly and reduces the coercive field (Nat Commun 11, 2836 (2020)). Past work has shown that above ~18% La, a nonpolar antiferroelectric phase emerges (ACS Applied Materials & Interfaces 10, 14914 (2018)). Therefore, our interest is to still maintain the ferroelectric phase, but manipulate the polarization magnitude / direction and the resultant domain structure such that a single-domained ferroelectric can be obtained by the application of an electric field. The relevant text is added in the revised manuscript on page 3.

Q # 4. *Why does the direction of the spontaneous polarization alter upon La doping on BiFeO₃?*

Response: The fundamental origins of the spontaneous polarization (its magnitude and direction) arise from the electronic structure of the Bi-ion in BiFeO₃. As such, ~90% of the spontaneous polarization arises from the 6s electrons in the Bi-ion (PHYSICAL REVIEW B 71, 014113 (2005)). Upon substituting Bi with La, one is effectively progressively removing the 6s electrons and thus the polarizability of the La-BiFeO₃ is progressively decreased. Indeed, beyond ~18% of La substitution, the material undergoes a phase transition from a polar state to an antipolar phase. Our prior work (Nat Commun 11, 2836 (2020)) as well as other publications (Advanced Functional Materials 20 (7), 1108-1115 (2010)) have studied the effect of rare earth substitution on the magnitude and direction of the polar vector. We have added this to the text in the revised manuscript on page 3.

Q # 5. *Authors have studied the inverse piezoelectric effect which is accompanied by the relationship between mechanical deformation in materials subjected to an applied electric field. Can authors comment on the amount of pressure or strain that has been formed inside the compounds?*

Response: We thank the reviewer for this interesting comment. In our experiment, we are not studying the strain effect, however it is ~0.2% in La-BiFeO₃ under 20kV/cm [J. Appl. Phys. 104, 104115 (2008)].

Q # 6. *Why in-plane poling leads to the formation of a single ferroelectric domain and why this is the novel feature for La-substituted BiFeO₃?*

Response: We thank the reviewer for this comment. The multivariant to single variant switching in ferroelectric is well demonstrated in our previous work [Advanced Materials 35, 2301934 (2023)], where the local and global domain can be formed based on the direction of the net local polarization and the electric field. As discussed in Figure 3 and Supplementary Figure 7 and 8, a single domain can be formed based on the local (domain) polarization orientation with respect to the external electric field. However, due to the La-substitution it has more possible switching pathways than the BiFeO₃ [Nat Commun 11, 2836 (2020)], which makes a single domain possible. This effect is anisotropic due to the substrate strain anisotropy [Supplementary Note 1] consistent with the previous observations, Advanced Materials 35, 2301934 (2023). A relevant text is added in the revised supplementary and manuscript.

Q # 7. In Figure 2, it will be Bi0.75 instead of Bi0.85. Authors should correct it.

Response: We thank the reviewer for pointing out the mistake which is now corrected in the revised manuscript.

Q # 8. Authors have studied the inverse spin Hall voltage. Can authors comment on the estimation of spin Hall conductivity associated with the spin current?

Response: We thank the reviewer for this comment. Since we are measuring the open circuit non-local voltage as a result of spin-charge conversion. To calculate the spin Hall conductivity, we require the absolute value of spin Hall angle and using, $\sigma_S = \sigma_{dc} \times \theta_S^{HA}$ where θ_S^{HA} , and σ_{dc} correspond to the spin Hall angle and the charge conductivity, respectively. Using the existing non-local geometry, we can estimate the SHA using angle dependent measurement in the presence of a magnetic field, which is out-of-scope of this work.

Q # 9. The magnitude of electric field required to switch the magnon spin current is found to be significantly smaller for a doped system in contrast to the parent compound. Why?

Response: BiFeO₃ exhibits rhombohedral distortion in the R3c space group having four possible ferroelastic/ferroelectric variants with the polarization along [111]_{pc} whereas La-doping modifies the rhombohedral distortion and hence tilts the polar vector away from [111]_{pc}. This is expected to open 12 possible switching paths in contrast to the 4 directions in pure BiFeO₃, which helps in reducing the switching energy landscape [Nat Commun 11, 2836 (2020)]. We have added the relevant text to the revised manuscript.

Q # 10. Authors should provide the effective values of second-order nonlinear susceptibility for both bare and doped systems through SHG method. Why did authors not implement the third harmonic generation for in-plane polarization mapping?

Response: Measurement of the effective values of the second-order nonlinear susceptibility would require measurement of absorption and usage of a reference crystal with known non-linear susceptibility elements [Appl. Phys. Lett. 92, 121915 (2008)]. Given that our measurement setup is only sensitive to in-plane symmetry, we do not believe there will be any difference between La-doped BiFeO₃ and BiFeO₃ as they have the same in-plane orientation, the only difference is in polarization magnitude which would simply reduce the values of the tensor elements.

In regard to second vs. third harmonic generation: Second harmonic generation (SHG) is a non-linear effect that occurs in all non-centrosymmetric crystals. Given that ferroelectrics are non-centrosymmetric all ferroelectrics will display SHG effects and based on the polarization dependence of the SHG intensity one can understand the underlying symmetry of the materials, therefore giving information about polarization direction and domain structure. SHG is a standard optical technique, used widely for probing ferroelectric systems [J. Am. Ceram. Soc., 94: 2699-2727 (2011)]. On the other hand, third-harmonic generation occurs in all systems, even air, and occurs with orders of magnitude less efficiency than SHG [Kajzar, F. "Third harmonic generation." Characterization techniques and tabulations for organic nonlinear optical materials. Routledge, 2018. 783-856 and Photonics 11(4), 313 (2024)]. Given the known sensitivity of SHG to ferroelectric polarization, we find that there is no additional benefit to the measurement of THG as it would require major modifications to the optical setup to detect such a signal. Currently, we have not found any study demonstrating in-plane polarization mapping with THG, whereas, SHG has been used numerous times over decades for imaging of ferroelectric domains [J. Am. Ceram. Soc., 94: 2699-2727 (2011)].

Q # 11. Why did authors use DyScO3 substrates for film deposition?

Response: DyScO₃ substrate has a good lattice match (~-0.3%) that helps in the epitaxial BiFeO₃ thin film growth. The relevant text is added in the method section.

Q # 12. Authors have considered the magnetic moment of Fe ion only in the Monte Carlo simulations, why? As the La ion also exhibits some magnetic moment in the doped system.

Response: La-replaces the Bi sites that contribute to the polarization. One electron in d-shell is not expected to contribute to the magnetization in comparison to Fe which is ~4.8 μ B. Also, in the related rare earth orthoferrite class of materials (which have been extensively studied in the past), the rare earth site orders magnetically at very low temperatures (~5K and below) and thus we believe the La site does not

contribute significantly to the net magnetic moment. Having said that, we do appreciate this question and some future theoretical studies are planned to address this.

Small contribution

Q # 13. Authors should provide some realistic names of future applications at the end of conclusion/summary.

Response: We thank the reviewer for the suggestions, and future applications are added to the summary /conclusions.

Q # 14. Authors should provide the full form of terms like DMI, SHG etc. when they appear first in the manuscript. Authors must carefully check the Supplementary Note numbers like 3 instead of 2 for “Ferroelectric Domain Characteristics”, 4 instead of 3 for “Optical second harmonic generation for in-plane polarization mapping (SHG)” etc. before resubmission.

Response: We apologize for leaving the inadvertent mistakes which are now corrected in the revised version.

Response to comments from Reviewer #4

Remarks: *In this work, the authors report on the demonstration of control of magnon transport with electric field in La-substituted BFO. This material offers enhanced properties compared to regular BFO, as it can be prepared in a single ferroelectric domain state with in-plane electrodes and produces a stronger magnon signal. I found that the paper is rather well-written, one can follow the arguments without too much difficulty, and plenty of details are provided in the Methods and supplement. This is an interesting spin transport experiment, but in my opinion, this is mainly an improvement of what is reported in refs. 30, 31 and 35 (from the same authors) and is therefore not suitable for publication in Nature Communications but rather in a more specialized journal. I also regret that not much explication of the mechanism of the observed effect is provided and the reason why it is so much more efficient in this material is not really given. More precisely, ref. 35 shows similar magnon transport experiments than here, but realized on regular BFO, so with stripy ferroelectric domains and a smaller non-reciprocity observed, while ref. 30 focuses on the switching of ferroelectric domains in regular BFO using electrodes and ref. 31 achieves also magnon transport between electrodes with a single cycloid wavevector in the domain in-between the electrodes by tuning the strain applied on the film with the substrate. In ref 31. similar results as here are obtained regarding the effect of the direction of k . In addition, I have numerous remarks and questions about the article, which I provide below to help the authors improve their manuscript.*

Response: We thank the reviewer for finding our experiments interesting and easily understandable. We appreciate the reviewer's constructive comments to improve the manuscript. Having said that, we respectfully disagree on the rest of this comment. Indeed, we have used similar measurement methods (which are also consistent with other such measurements by peers in the field), The MAIN focus of our paper is to understand the nature of spin/magnon transport in a sample that has a single ferroelectric domain. Past work by us and other workers in the field (PRL 129, 087601 (2022),) has all focused on materials that have ensembles of ferroelectric domains. The existence of such ferroelectric domains is inevitably a cause for the scattering of spins during their transport (PRL 129, 087601 (2022)). Thus, our main goal in this work was to use the La substitution to tune the crystal structure, the spontaneous polarization and thus the ferroelectric domain structure such that it evolves easily into a single-domained ferroelectric state. The substrate structural anisotropy plays a key role in this as well. Once a single-domained structure is created, we can then explore the spin cycloid and spin/magnon transport thru the system. This is the main focus of our paper. Regarding ref.31, that work is complementary to what is reported in this paper, but the underlying physics is not the same. The use of La to manipulate the polarization, the domain structure, and thus its switchability is completely unique in this paper, which does not play ANY role in ref.31.

We are grateful to the reviewer for constructive comments to improve the manuscript. Here we provide a point-by-point response to the comments and suggestions. We hope that the revised manuscript qualifies for acceptance.

Figures

There is in my opinion a general lack of rigor and clarity in the design of the figures (in main text and extended data). None of the STEM, SHG, PFM or NV maps has a colorscale specifying the displayed quantity and its values. This should be corrected. For the PFM, what is the probed polarization direction? For the NV data, one has to reach the very end and the Methods section to finally learn that this is iso-B data! This should be specified on the figures or in the captions!

Response: We thank the reviewer for the opinion. We note that the color scales are not critical for understanding the qualitative comparison in PFM, which is what we are aiming for. Moreover, NV and SHG and STEM color scales are already reproduced in the supplementary for clarity. As per suggestions, we have now added all the information to the maps and figure captions.

Fig 1:- Panel a: this plot is very strange to me, and I have the impression that it could actually be replaced by a table and become more readable. What are the dashed lines? Why use different symbols for the different states (which are anyway placed at different positions along the x axis) rather than having one symbol for BFO and another one for BLFO? To which state on the graph does the "complex state" label refer?

Response: We have re-plotted the figure as well as supplied the table in the method section. All other notations such as lines and symbols are fixed.

- Panel b: I do not understand this figure. What is supposed to be varied on the x axis? Is it implied here that there would be 2 phases in BFO with different rotational sense of the cycloid? This rotational sense is fixed by the magnetoelectric DMI-term, so it is linked to P, which does not appear on the figure. In addition, the depiction of the "complex state" is not very clear. I think that this panel is not necessary.

Response: Generally, such a figure as in (b) aims to show the energy landscape of a ferroelectric. Due to the bistability of the polar state, the double well structure (which is captured SCHEMATICALLY in this figure) shows the depth of the double well as a function of the order parameter. Under normal circumstances, this would have the total energy on the y-axis and the magnitude of the order parameter, Polarization, on the x-axis. Here we are simply aiming to compare BiFeO₃ with La-BiFeO₃ on a qualitative level. We have now added the x-axis as well to fig (b).

- Panels c-d: I suggest to specify that the blue-green arrows represent P.

Response: We thank the reviewer for the suggestion. We have mentioned the colors of the arrows.

- Panel f: I am surprised by the curved wavefront of the cycloid on this image, usually BFO/DSO imaged with iso-B mode provides an undistorted zigzag pattern with straight lines and a 90° angle at the ferroelectric domain walls, which does not seem to be the case here. Can the authors comment on this? Is there thermal drift in their system despite the use of iso-B mode?

Response: We have never observed a sharp 90 cycloid in any of our sample/measurements (Nat Commun 15, 2903 (2024)), which seems consistent with other groups who have reported on the imaging of cycloids [Nature 549, 252–256 (2017), Nat Commun 15, 1902 (2024), PHYS. REV. APPLIED 17, 044051 (2022), Zoomed Figure 3e]. We believe this is simply due to the resolution limit of the NV microscope. Similar features are also reported on the Qnami Application note Qnami_AppNote1_BFO.pdf. We have cited references from other expert groups (PHYS. REV. APPLIED 17, 044051 (2022)) for better understanding. We have made sure that there is no thermal drift effect in our measurements. These references are further specified in the text in the revised manuscript.

- Panel h: Why are the uniform looking patches designated as non-collinear phase in the image? The caption mentions a canted AFM phase, which seems right, but this is not a non-collinear phase.

Response: We agree with the reviewer and figure text has been corrected in the revised manuscript.

Fig 2:

- Do not use P both for polarization and injected power.

Response: We thank the reviewer for the suggestion, which is now rectified in the revised manuscript.

- Along which direction is P in panel b?

Response: The direction of P and E are added in the revised figure.

- Panel e: What is the width of the gap between the electrodes for the BFO data? In the caption, domains are mentioned concerning BFO, I guess that the authors mean ferroelectric domains but this should be precised.

Response: As suggested, the caption is revised with all information.

Fig. 3:

Why is the contrast so much weaker in the NV data in the column d,j,q? The cycloid period also looks different between both cases, especially in panels c and d, why? And I honestly cannot really see a cycloid in panel j, the data quality is not good enough.

Response: The difference in the pixel quality is expected due to ex-situ electric field poling and measured at different available time schedules however features are distinguishable for a single and multidomain state. Also, tip quality can be compromised with time. The image quality in Figure J is relatively less, however multidomain state is still visible. The picture quality might be compromised during the submission process. We are providing separate high-resolution figures with the revised manuscript. The period of the cycloid is the same but looks different due to the different scale of the measurements. The scale bars are different in different devices. We have added the text in the caption to highlight this concern. We thank the reviewer for important comments.

Fig 4: I find panels b, c, d a little difficult to understand.

Response: We thank the reviewer for an important concern. A further explanation is added in the figure caption and the text.

Ext. Data Fig. 2: In all cases except g-h, there are always two directions of P present between the electrodes, so I do not understand what the P arrows mean.

Response: We appreciate the reviewer for this concern. The arrow P means the net P direction within the device without any electric field a,b and e,f and with electric field c,d and e,h. However we agree on the part that figures a,b, and e,f are the data without field which means there is no preferred direction. We can therefore say the direction of P is decided by the local polarization. We have added more text and modified the figure presentation.

Ext. Data Fig. 4: What is Vnl?

Response: Vnl represents the non-local voltage which is a typo. We have corrected the representation of ISHE voltage in the revised manuscript.

Text

Could the authors give more insight about why La-substitution changes the direction of P, and why including 15% of La decreases P by 50%? And why was 15% chosen? Is it actually an optimal value or chosen because it was experimentally convenient?

Response: It is well established that the 15% La doping impacts the energy landscape significantly and reduces the coercive field to a minimum value (Nat Commun 11, 2836 (2020)). Past work has shown that above ~18% La, a nonpolar antiferroelectric phase emerges (ACS Applied Materials & Interfaces 10, 14914 (2018)). Therefore, our interest is to still maintain the ferroelectric phase, but manipulate the polarization magnitude and the resultant domain structure such that a single-domain ferroelectric can be obtained by the application of an electric field. The relevant text is added in the revised manuscript on page 3.

- End of 1st column, page 3: I do not understand. P is decreasing, thus reducing DMI, which is responsible for the stabilization of the cycloid and the tilt corresponding to the spin density wave. Why would this tilt increase when P decreases? Should it not decrease too? And if there is actually an increase, it might be possible to detect it with scanning NV magnetometry. The field measured, which originates from the spin density wave, should be larger in BLFO in this case. Such data would be interesting.

Response: We appreciate the reviewer for bringing up very interesting point, which was missed in the first draft. We agree on the point that a larger tilt will introduce a higher magnetic field in NV measurements when the full-B is measured, as discussed in the supplementary Figure 16. In the pristine state, the total field is $\sim \pm 5\text{G}$ with the canted AFM phase which changes to $\sim \pm 1\text{G}$ after poling where only the spin cycloid is present. We have explicitly mentioned it in the text. We once again thank the reviewer for adding an interesting comment.

Side note: the reference to the polar maps in Fig 1 here should be the panels c and d, rather than a and b.

Response: We thank the reviewer for noting this. These are fixed.

- End of 2nd column, page 3: "a single, in-plane pulse leads to the deterministic switching and formation of a single multiferroic domain": the magnetic state between the electrodes has not been mentioned in the text yet, so claiming that there is a single multiferroic domain sounds a little premature.

Response: We thank the reviewer for noting this. It is not fixed in the revised manuscript.

- 1st column, page 5: why is the signal so much larger (by a factor 4?) in BLFO compared to BFO? And if we compare with the experiment on TSO with a single k direction in ref 31, is it also so much larger?

Response: The experiment in ref. 31 was done on a different substrate. The measurement strategy is also different. Those measurements were done without electrical poling to understand the physics of cycloid anisotropy. That magnitude also involved the equal offset which was not considered therefore the magnitude comparison cannot be done here.

In our case, the data was recorded on BiFeO₃ and La- BiFeO₃ using an identical scheme, where we discussed a single domain ferroelectric vs. multidomain system.

- End of 1st column, page 6: a reference to Fig. 4 is missing.

Response: Reference is added.

- 2nd column, page 6: *"topology of the spin cycloid": the authors should refrain from using the word topology here, this could be misleading as a cycloid is a topologically trivial state.*

Response: As per the reviewer's suggestion, the irrelevant text is removed.

- 2nd column, page 6: *"uniform magnetic texture": an antiferromagnetic cycloid with an attached spin density wave is a very complex non-collinear magnetic state, not a uniform state!*

Response: These changes have been made.

- 2nd column, page 6: *I think that what would lead to a null magnon signal is a balanced combination of domains. Otherwise, if a domain with a given cycloidal state is dominant, would not the signal only be reduced?-* Beginning of 1st column, page 7: *"This can be understood from the symmetry of P and k , if we consider that La -substitution can allow for different symmetry operations when switching the polarization." Can the authors be more specific here?*

Response: Thanks for the comments. This text corresponds to the devices with finite signals with single-domain features. We also agree with the reviewer on what leads to the null signal as a response to the balanced domain. The text is further elaborated and as suggested the origin of the null signal is also added in the revised manuscript.

- 1st column, page 7: *"This elucidates the anisotropic nature of magnon transport as it is intricately linked to the spin cycloid and thus the polarization of the BLFO": this is an overstatement. The anisotropic magnon transport is experimentally shown here, but not microscopically explained. And actually, it would have been extremely surprising if the magnetic cycloid had no effect on the spin waves.*

Response: We agree with the reviewer that the spin cycloid indeed affects the spin waves. This is what we have presented in Figures 3 and 4 where the different orientation of the magnetic cycloid gives different inverse spin Hall voltage outputs. As suggested by the reviewer, we have revised and added relevant text in the revised manuscript.

Supplementary

- p 26: *in dual iso-B mode, 2 microwave frequencies are used, to get rid of the artifacts that could come from PL variation unrelated to the magnetic state.*

Response: We agree with the reviewer and as suggested, the text is revised in the supplementary.

- Suppl. Fig. 15: *Why is the cycloid not visible in panel c? The bias field contribution should be subtracted from panel f.*

Response: It is probably a consequence of the file conversion issue. It is quite evident in our files. We have tried to increase the resolution in the revised version. We hope that cycloids are visible in the revised version. We have also subtracted the bias field contributions from the full-B data.

We are grateful to the reviewer for very useful comments and suggestions which improved the manuscript to par level.

Reviewer #5 (Remarks to the Author):

Response: We thank the reviewer for providing the input on our manuscript.

REVIEWER COMMENTS

Reviewer #1 (Remarks to the Author):

I appreciate the authors' efforts to responding to the reviewers' comments with additional experiments and explanations. Because the article title has been modified, the first harmonic signal is not related to the thermally excited magnon current, thus the figure is not necessary to be added into Supplementary Materials. This response alleviates my concerns and the revised manuscript is much improved. Therefore, I recommend this manuscript for publication in Nature Communications.

Reviewer #2 (Remarks to the Author):

Authors have taken care all my concerns meticulously. The paper can now be published.

Reviewer #3 (Remarks to the Author):

I thank the authors for their response to my comments and to those of the other reviewers. The manuscript has been significantly improved. However, I still think that this paper would be more suitable for a specialized journal than Nature Communications. I am not fully convinced by the arguments of the authors about the novelty of the work, as both the effects of La substitution in BFO and the principle of the thermal magnon transport experiments on BFO are already reported in the literature. Furthermore, despite the significant amount of data presented both in the main text and in the extensive Supplementary Information, not much is given about the underlying mechanism for the observed effect.

Concerning the answers to my comments, I strongly disagree with the authors, colorbars are not useless. In particular, I have been misled by the implied use of the same color scale for all NV images in Fig 3. I assumed that all the data had been measured in the same conditions with the same tip and thus that the contrast could be compared, which is not the case.

Still about Fig. 3, I am sorry but even with the high resolution images I can't see a cycloid in panel j. In addition, I can't tell by eye if the apparent period difference really agrees with the different scale of the images. The authors should have provided line profiles extracted from the panels c-d, i-j and p-q.

I also still do not understand why reducing P enhances the tilting. I agree that the data shown in Fig. S17 (there is a mistake in the numbering in the text) supports this, but I don't get why if the cycloid is less stable it means that the tilt should increase.

I do not see a single cycloid variant between the electrodes in Fig S18 a-b, although it is stated that a single multiferroic domain is obtained in the [1–10] devices.

Reviewer #4 (Remarks to the Author):

I co-reviewed this manuscript with one of the reviewers who provided the listed reports. This

is part of the Nature Communications initiative to facilitate training in peer review and to provide appropriate recognition for Early Career Researchers who co-review manuscripts.

Response to comments from Reviewer#1

Reviewer #1 (Remarks to the Author): I appreciate the authors' efforts to responding to the reviewers' comments with additional experiments and explanations. Because the article title has been modified the first harmonic signal is not related to the thermally excited magnon current, thus the figure is not necessary to be added into Supplementary Materials. This reponse alleviates my concerns and the revised manuscript is much improved. Therefore, I recommend this manuscript for publication in Nature Communications.

Response: We are grateful to the reviewer for appreciating the revised manuscript and recommending the it for publication.

Response to comments from Reviewer#2

Reviewer #2 (Remarks to the Author): Authors have taken care all my concerns meticulously. The paper can now be published.

Response: We appreciate the reviewer for considering the revised version and accepting it for publication.

Response to comments from Reviewer#3

Reviewer #3 (Remarks to the Author): I thank the authors for their response to my comments and to those of the other reviewers. The manuscript has been significantly improved. However, I still think that this paper would be more suitable for a specialized journal than Nature Communications. I am not fully convinced by the arguments of the authors about the novelty of the work, as both the effects of La substitution in BFO and the principle of the thermal magnon transport experiments on BFO are already reported in the literature. Furthermore, despite the significant amount of data presented both in the main text and in the extensive Supplementary Information, not much is given about the underlying mechanism for the observed effect.

Response: We thank the reviewer for appreciating our revised manuscript. However, we respectfully do not agree with the comment about “already reported on the magnon transport of La-BFO”. Thermal magnon transport in pure BFO has been studied (Parsonnet et al, PRL 125, 067601 (2022)); however, La-substitution produces substantive changes to the electronic structure, polarization and the domain structure. As such, to the best of our knowledge, we have not seen any reported work on magnon transport in La-doped BFO particularly single domain of BFO. Understanding magnon transport in a single ferroelectric domain is fundamentally important since one is able to study spin transport without the role of ferroelectric domain wall scattering. In some sense, this is akin to carrying out measurements on single crystals. Having said that, we would appreciate it if the reviewer can enlighten us with the relevant published literature on both of the novel concepts proposed in this work that have been demonstrated, that we might have missed. We would be very happy to cite them.

We next provide point-by-point responses to the second round of comments.

Comment#1: Concerning the answers to my comments, I strongly disagree with the authors, colorbars are not useless. In particular, I have been misled by the implied use of the same color scale for all NV images in Fig 3. I assumed that all the data had been measured in the same conditions with the same tip and thus that the contrast could be compared, which is not the case.

Still about Fig. 3, I am sorry but even with the high resolution images I can't see a cycloid in panel j. In addition, I can't tell by eye if the apparent period difference really agrees with

the different scale of the images. The authors should have provided line profiles extracted from the panels c-d, i-j and p-q.

Response: The color bars in the figures have been added in the revised manuscript. As per suggestion, we have also added a figure corresponding to the line scans on the NV data of figure 3 c-d, i-j, and p-q as supplementary Figure. It shows that the period does not change appreciably after electric field switching. We have included this figure in the supplementary.

Figure: Line scan on the NV data corresponding to device a,d,g [010], b,e,h [100] and c,f,i [1-10] (from main text Figure 3c-d, i-j, p-q). Top and Bottom NV images are corresponding to the left and right poled configurations.

Comment#2: I also still do not understand why reducing P enhances the tilting. I agree that the data shown in Fig. S17 (there is a mistake in the numbering in the text) supports this, but I don't get why if the cycloid is less stable it means that the tilt should increase. I do not see a single cycloid variant between the electrodes in Fig S18 a-b, although it is stated that a single multiferroic domain is obtained in the [1-10] devices.

Response: (i) The device orientation was inadvertently written as [1-10]. It belongs to [100] which is now corrected in the revised manuscript.

(ii) Regarding the tilt vs polarization: The coupling between octahedral tilt and spontaneous polarization with and without La-substitution has been studied in multiple publications, the most recent being an ab initio study, by Iniguez et al, PHYSICAL

REVIEWB106,165122 (2022) and experimentally in Adv. Mat. 23, 1765–1769 (2011), Adv. Funct. Mat. 20, 1108–1115 (2010) and Nat. Comm. 11,2836 (2020). We have made edits to main text to explicitly describe the conclusions from PHYSICAL REVIEWB106,165122 (2022).

Reviewer #4 (Remarks to the Author): I co-reviewed this manuscript with one of the reviewers who provided the listed reports. This is part of the Nature Communications initiative to facilitate training in peer review and to provide appropriate recognition for Early Career Researchers who co-review manuscripts.

Response: We thank the reviewer for reading the manuscript and other reviewers reports.

REVIEWERS' COMMENTS

Reviewer #3 (Remarks to the Author):

I thank the authors for their answer and for providing the line profiles across the cycloid. I do not have any further questions or comments about the data presented and its interpretation. Following the reports of the other referees, I do not oppose to the publication of this work in Nature Communications.